# Enhancing Visual Prompting through Expanded Transformation Space and Overfitting Mitigation

**Shohei Enomoto**
NTT
Tokyo, Japan
`shohei.enomoto@ntt.com`

## Abstract

Visual prompting (VP) has emerged as a promising parameter-efficient fine-tuning approach for adapting pre-trained vision models to downstream tasks without modifying model parameters. Despite offering advantages like negligible computational overhead and compatibility with black-box models, conventional VP methods typically achieve lower accuracy than other adaptation approaches. Our analysis reveals two critical limitations: the restricted expressivity of simple additive transformation and a tendency toward overfitting when the parameter count increases. To address these challenges, we propose ACAVP (Affine, Color, and Additive Visual Prompting), which enhances VP's expressive power by introducing complementary transformation operations: affine transformation for creating task-specific prompt regions while preserving original image information, and color transformation for emphasizing task-relevant visual features. Additionally, we identify that overfitting is a critical issue in VP training and introduce TrivialAugment as an effective data augmentation, which not only benefits our approach but also significantly improves existing VP methods, with performance gains of up to 12 percentage points on certain datasets. This demonstrates that appropriate data augmentation is universally beneficial for VP training. Extensive experiments across twelve diverse image classification datasets with two different model architectures demonstrate that ACAVP achieves state-of-the-art accuracy among VP methods, surpasses linear probing in average accuracy, and exhibits superior robustness to distribution shifts, all while maintaining minimal computational overhead during inference. Our code is available at `https://github.com/s-enmt/ACAVP`.

## 1 Introduction

Visual prompting (VP) [Bahng et al., 2022, Elsayed et al., 2019] has emerged as a promising parameter-efficient fine-tuning technique for adapting pre-trained vision models to downstream tasks. By learning task-specific noise applied to input images while keeping the model parameters frozen, VP offers significant advantages: it introduces negligible computational overhead and can be applied to black-box models where internal parameters are inaccessible Tsai et al. [2020], Oh et al. [2023].

Despite these advantages, VP methods typically achieve lower accuracy than full fine-tuning [Bahng et al., 2022]. A straightforward approach to improving VP accuracy is to increase the size of the prompt region, thereby enhancing the expressive power through increased parameterization. Theoretically, since the transformation space with increased parameters becomes a superset of the original transformation space, the approximation error should decrease with more parameters [Cai et al., 2024b]. However, as our preliminary experiments show, this approach is ineffective. As illustrated in Table 1, increasing the number of VP parameters does not improve training accuracy,

39th Conference on Neural Information Processing Systems (NeurIPS 2025).

and more importantly, leads to lower test accuracy, indicating overfitting issues rather than enhanced expressive power.

Table 1: Results of preliminary experiments comparing performance between padding-type noise addition and image-sized noise addition.

| Image | | |
|---|---|---|
| Params | 23,280 | 50,176 |
| Train Acc | $97.53_{\pm 0.14}$ | $97.35_{\pm 0.18}$ |
| Test Acc | $93.65_{\pm 0.04}$ | $89.52_{\pm 0.12}$ |

Our analysis reveals two critical limitations in VP: (1) simply adding more parameters with the traditional additive transformation has inherent limitations in its expressive power, and (2) increasing parameter count leads to overfitting issues. This leads us to a key question. How can we enhance the expressive power of VP while effectively preventing overfitting?

To address this challenge, we propose ACAVP (Affine, Color, and Additive Visual Prompting), a novel VP transformation design that enhances accuracy while maintaining minimal computational overhead. ACAVP incorporates two parameterized image transformation techniques into existing VP methods: affine and color transformations. The affine transformation component generates VP regions through operations such as resizing and rotation while preserving the original image information. In contrast, the color transformation component enhances the original image information through adjustments to brightness and contrast. Importantly, both components retain the advantages of VP technology because of their negligible computational cost compared to model inference. Furthermore, to address the overfitting problem, we comprehensively evaluated various regularization techniques for VP training. After experimenting with several strategies, including dropout, MSE loss regularization, and weight decay, we found that data augmentation methods, particularly TrivialAugment [Müller and Hutter, 2021], are most effective for mitigating overfitting in VP training. TrivialAugment provides a low-cost yet powerful approach to increase input diversity during training, significantly stabilizing the learning process and improving generalization performance.

We evaluated ACAVP's effectiveness across twelve image classification datasets using two different architectures. Our method outperforms existing VP approaches and achieves higher average accuracy than linear probing. Additionally, ACAVP demonstrates superior out-of-distribution (OOD) performance due to its improved generalization capabilities. The computational overhead of ACAVP is negligible compared to the inference time of the recognition model.

Our main contributions are as follows:

- We propose ACAVP, a novel VP method that enhances the expressive power of visual prompting by introducing complementary transformation operations: affine transformation for creating task-specific prompt regions while preserving original image information, and color transformation for emphasizing task-relevant visual features. This approach achieves state-of-the-art accuracy among VP methods while maintaining minimal computational overhead.

- We identify and address the critical overfitting problem in VP by empirically evaluating regularization techniques. We demonstrate that TrivialAugment effectively mitigates overfitting in ACAVP and existing VP methods, providing a general technique for improving VP performance.

- We evaluated our approach through extensive experiments across twelve diverse image classification datasets and two different model architectures. Our experiments demonstrate that ACAVP achieves the highest average accuracy among VP methods, surpasses linear probing in average accuracy, and exhibits superior robustness to distribution shifts in out-of-distribution scenarios.

## 2   Related Work

Visual prompting (VP) [Bahng et al., 2022] has emerged as a promising parameter-efficient fine-tuning paradigm. First introduced as adversarial reprogramming (AR) by Elsayed et al. [2019], VP adapts pre-trained models to downstream tasks by applying learnable prompts to input images while keeping the model parameters frozen. This approach enables efficient adaptation without modifying the underlying model architecture.

**Transformation design for VP.** Although the original VP adds padding-type noise to the image, several studies have focused on improving its transformation design. Wu et al. [2024] introduced enhanced visual prompting (EVP), which preserves original image information by resizing the image and adding prompts around it. Building on this concept, Tsao et al. [2024] proposed AutoVP, an end-to-end framework that automates VP design choices and introduces learnable input scaling parameters to optimize the resize intensity dynamically. In a different direction, Jin et al. [2025] developed Low-Rank matrix multiplication for visual prompting (LoR-VP), which uses low-rank matrix decomposition to ensure that visual prompts influence all image patches while considering induced biases between patches. Rather than using a single prompt for an entire task, several approaches generate customized prompts for individual samples. Oh et al. [2023] proposed Coordinator, which uses an encoder-decoder network to generate sample-specific prompts. Similarly, Cai et al. [2024b] proposed sample-specific multi-channel masks (SMM) that generate appropriate prompt regions for each individual sample. Cho et al. [2024] expanded this concept by generating prompts in both spatial and frequency domains using a network for sample-specific adaptation. While sample-specific VP approaches achieve high expressive power, they significantly increase inference-time computational overhead, undermining one of VP's primary advantages. In contrast, our work focuses on task-level visual prompting while addressing its limitations.

**Learning strategies for VP.** Researchers have also explored improved learning strategies for VP. Wu et al. [2024] incorporated techniques from adversarial attacks, specifically input diversity [Xie et al., 2019] and gradient normalization [Goodfellow et al., 2015], to improve training stability. Cai et al. [2025] proposed attribute-based visual reprogramming (AttrVR), leveraging descriptive and distinctive attributes to mitigate ambiguities caused by fixed template prompts and promote more context-aware alignment between images and attributes. Zhang et al. [2024] introduced transferable visual prompting (TVP), which learns a single VP to improve performance across diverse multimodal large language models. In the non-CLIP model, a random class mapping is used to align the pre-trained model classes with the downstream tasks' classes. Wu et al. [2024] proposed a method that first infers the pre-trained model on a downstream dataset and then performs label mapping according to its output distribution. Chen et al. [2023] introduced iterative label mapping (ILM), which automatically remaps source labels to target labels, progressively improving VP performance on target tasks. Taking a probabilistic approach, Cai et al. [2024a] proposed bayesian-guided label mapping (BLM), which constructs a sequentially updated stochastic label mapping matrix quantifying pairwise relationships between pre-trained labels and downstream labels. In this study, we show that VP training is less accurate due to overfitting, and that countermeasures, especially data augmentation, are the most effective learning strategy.

We propose a novel VP design that enhances expressive power while minimizing the computational cost increase in inference time. Additionally, we introduce data augmentation techniques to mitigate overfitting, a common issue in VPs. Our approach balances expressive power and computational efficiency, maintaining VP's key advantages while improving performance.

## 3  Proposed Method: ACAVP

We propose a novel VP approach, ACAVP (Affine, Color, and Additive Visual Prompting), to enhance VP by introducing additional transformation operations that address its limitations. Our investigation begins with identifying the primary constraints of existing VP techniques and then proposing solutions that maintain the low computational overhead characteristic of VP methods. Figure 1 shows an overview of ACAVP. The theoretical justification for our approach is provided in Appendix B, where we formally analyze the hypothesis spaces of existing VP methods and ACAVP, demonstrating that ACAVP has a smaller approximation error, which explains its superior performance. The implementation details are provided in Appendix C.

### 3.1  Preliminaries

VP is a parameter-efficient fine-tuning technique that adapts pre-trained recognition models to downstream tasks by applying learnable prompts to input images while keeping the model parameters frozen [Bahng et al., 2022]. During training, VP learns a prompt that minimizes the task-specific loss of the recognition model. At inference time, the learned prompt is applied to input images before model prediction. Formally, given an input image $x \in \mathbb{R}^{3 \times H \times W}$, traditional VP transforms

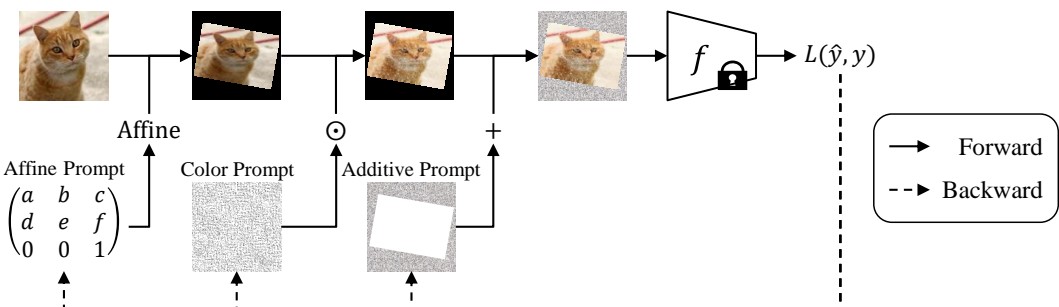

Figure 1: Overview of ACAVP. The input image undergoes three transformations: first, an affine transformation using the affine prompt, followed by a color transformation using the color prompt, and finally an additive transformation using the additive prompt. The resulting VP image is fed into a frozen recognition model, and the loss is backpropagated to update all three prompt parameters.

it into a prompted image $\tilde{x} \in \mathbb{R}^{3 \times H \times W}$ using the following equation: $\tilde{x} = x + M \odot \delta$, where $M \in \{0, 1\}^{H \times W}$ is a binary mask indicating the prompt region, $\delta \in \mathbb{R}^{3 \times H \times W}$ is the learnable prompt parameter, and $\odot$ denotes element-wise multiplication.

VP has advantages such as almost zero computational overhead during inference and applicability to black-box recognition models (including cloud APIs) [Oh et al., 2023, Tsai et al., 2020] in order to maintain the original model architecture and parameters. However, as our preliminary experiments show, VP suffers from two significant limitations: limited expressive power due to its reliance on simple additive transformations and a tendency to overfit when the parameter count increases.

### 3.2 Problem Analysis

Our preliminary experiments revealed two significant limitations of conventional VP approaches. First, increasing the number of VP parameters by expanding the prompt size theoretically reduces the approximation error, but does not improve the test or training accuracy. This suggests that the additive transformation alone has inherent limitations in its expressive power. Second, as the number of parameters increases, the VP model tends to overfit, reducing test accuracy. Based on these observations, we hypothesized that: (1) introducing different types of transformations beyond addition could enhance the expressive power of VP, and (2) introducing effective regularization strategies could mitigate overfitting issues.

### 3.3 ACAVP Design

To address these limitations, we propose ACAVP (Affine, Color, and Additive Visual Prompting), an enhanced VP method incorporating affine transformation and color transformation into the VP design, accompanied by effective regularization through data augmentation techniques. Our proposed method transforms the input image according to:

$$\tilde{x} = \text{Affine}(x, A) \odot \hat{\sigma} + M \odot \delta \tag{1}$$

where $\text{Affine}(\cdot, A)$ represents the affine transformation with learnable affine prompt $A \in \mathbb{R}^{3 \times 3}$, $\hat{\sigma} \in \mathbb{R}^{3 \times H \times W}$ is the learnable color prompt, and $\delta \in \mathbb{R}^{3 \times H \times W}$ is the learnable additive prompt. Unlike conventional VP methods, $M \in \{0, 1\}^{H \times W}$ is dynamically generated after the affine transformation is applied, where $M$ has value 1 for pixels where all three channels are zero, and 0 elsewhere. This ensures the additive prompt is only applied to the empty regions that the affine transformation creates. This formulation extends the transformation space beyond simple addition, enabling more expressive prompts while maintaining the computational efficiency of VP.

**The affine transformation prompt** generates task-specific feature regions while preserving the original image information. The affine transformation enables geometric modifications, including rotation, translation, shearing, and scaling, providing a richer set of transformations than simple addition. For an input point $(u, v)$, the affine transformation with transformation matrix $A$ computes its new coordinates $(\hat{u}, \hat{v})$ as follows:

$$\begin{bmatrix} \hat{u} \\ \hat{v} \\ 1 \end{bmatrix} = \begin{bmatrix} \hat{s_x}\cos\hat{\theta} + s\hat{h}_x\sin\hat{\theta} & -\hat{s_x}\sin\hat{\theta} + s\hat{h}_x\cos\hat{\theta} & \hat{t_x} \\ \hat{s_y}\sin\hat{\theta} + s\hat{h}_y\cos\hat{\theta} & \hat{s_y}\cos\hat{\theta} + s\hat{h}_y\sin\hat{\theta} & \hat{t_y} \\ 0 & 0 & 1 \end{bmatrix} \begin{bmatrix} u \\ v \\ 1 \end{bmatrix} \qquad (2)$$

To ensure stable training and prevent extreme transformations that might destroy the original image information, we constrain the transformation parameters using activation functions: $\hat{t} = \tanh(t) \cdot R_t$, $\hat{\theta} = \tanh(\theta) \cdot R_\theta$, $\hat{s} = \text{sigmoid}(s)$, and $\hat{sh} = \tanh(sh) \cdot R_{sh}$. Here, $t$ represents learnable translation parameters, $\theta$ is a learnable rotation parameter, $s$ refers to learnable scaling parameters, and $sh$ denotes learnable shearing parameters. The hyperparameters $R_t$, $R_\theta$, and $R_{sh}$ control the maximum range of each transformation. The default values for these hyperparameters are set to $R_t = 0.05$, $R_\theta = R_{sh} = 0.1$.

**The color transformation prompt** further enhances the expressive power of our VP by adjusting image properties such as brightness and contrast. This is achieved through a learnable parameter $\sigma \in \mathbb{R}^{3 \times H \times W}$ applied as a multiplicative factor to the transformed image. The color transformation allows the prompt to emphasize or attenuate specific features in the original image, potentially highlighting task-relevant information.

To prevent excessive color transformation that might corrupt the original image information, we constrain the values using: $\hat{\sigma} = \text{sigmoid}(\sigma) \cdot R_\sigma$, where $R_\sigma$ is a hyperparameter controlling the maximum magnitude of color transformation, with a default value of $R_\sigma = 6.0$. This ensures that the color transformations remain within a reasonable range while still providing sufficient flexibility to enhance task-relevant features.

### 3.4 Overfitting Mitigation

As shown in our preliminary experiments, increasing the number of VP parameters theoretically reduces the approximation error, but causes overfitting. This observation led us to hypothesize that effective regularization strategies would be crucial for our enhanced ACAVP approach with its additional transformation parameters. To identify the most effective overfitting mitigation technique, we comprehensively evaluated various regularization approaches applied to VP training. Specifically, we investigated four categories of techniques: data augmentation, dropout applied after feature extraction layer, MSE loss between original and prompted images as an additional regularization loss term, and weight decay. Our experimental results, detailed in Section 4.4, revealed that data augmentation methods consistently provided the most substantial improvements in generalization performance. Among the various augmentation techniques evaluated, TrivialAugment emerged as particularly effective for VP training. TrivialAugment applies a randomly selected augmentation operation with random magnitude to each input image, providing sufficient diversity to mitigate overfitting while maintaining implementation simplicity. While previous VP approaches have employed basic augmentations like random crop and horizontal flip, we found that more sophisticated augmentation strategies could significantly improve the performance of VP.

## 4 Experiments

To evaluate the effectiveness of ACAVP, we conducted extensive experiments across 12 diverse image classification datasets using two different model architectures. We compared ACAVP against zero-shot inference (ZS) with CLIP [Radford et al., 2021] and state-of-the-art VP methods (VP [Bahng et al., 2022], EVP [Wu et al., 2024], and AutoVP [Tsao et al., 2024]). All experiments were repeated three times, and we report the average accuracy and standard error. Each dataset was split into training, validation, and test sets, and we used the checkpoint with the highest accuracy on the validation set for final testing. Detailed experimental settings are provided in Appendix D.

### 4.1 Main Results

We evaluated ACAVP using a ViT-B32-based CLIP model [Radford et al., 2021, Dosovitskiy et al., 2021] across 12 diverse image classification datasets. Table 2 shows the comparative performance results. ACAVP demonstrates superior accuracy on most datasets compared to existing VP methods, achieving an average accuracy of 83.18%, which significantly outperforms VP, EVP, and AutoVP. The

Table 2: Classification accuracy comparison on 12 image classification datasets using CLIP ViT-B/32. **Bold** and underlined values denote the first and second highest accuracy, respectively. The LP and FT results refer to Bahng et al. [2022], Tsao et al. [2024], Tang et al. [2024].

| Method | ZS | VP | EVP | AutoVP | ACAVP | LP | FT |
|---|---|---|---|---|---|---|---|
| CIFAR10 | 88.93 | 93.65 $\pm$ 0.04 | 95.87 $\pm$ 0.06 | 95.39 $\pm$ 0.06 | **96.27** $\pm$ 0.07 | 95.00 | 95.80 |
| CIFAR100 | 63.09 | 74.71 $\pm$ 0.09 | 78.33 $\pm$ 0.09 | 79.89 $\pm$ 0.04 | **80.06** $\pm$ 0.25 | 80.00 | 82.10 |
| CLEVR | 20.70 | 76.40 $\pm$ 0.56 | 76.82 $\pm$ 0.27 | **79.19** $\pm$ 0.29 | 77.19 $\pm$ 0.17 | 66.00 | 94.40 |
| DTD | 42.91 | 57.19 $\pm$ 0.14 | 60.89 $\pm$ 0.57 | 62.81 $\pm$ 0.12 | **67.43** $\pm$ 0.83 | 74.60 | 72.30 |
| EuroSAT | 39.02 | 96.36 $\pm$ 0.07 | 97.27 $\pm$ 0.08 | 97.32 $\pm$ 0.11 | **97.51** $\pm$ 0.09 | 95.30 | 97.90 |
| Flowers | 61.63 | 67.83 $\pm$ 0.47 | 77.76 $\pm$ 0.59 | 82.69 $\pm$ 0.80 | **89.36** $\pm$ 0.19 | 96.90 | 97.40 |
| Food | **79.67** | 78.92 $\pm$ 0.03 | 77.48 $\pm$ 0.00 | 76.53 $\pm$ 0.03 | 77.86 $\pm$ 0.09 | 84.60 | 80.50 |
| GTSRB | 18.18 | 91.38 $\pm$ 0.28 | 76.17 $\pm$ 2.40 | **92.50** $\pm$ 0.14 | 92.04 $\pm$ 0.38 | 85.80 | 98.90 |
| Pets | 85.96 | 86.51 $\pm$ 0.30 | 87.85 $\pm$ 0.06 | 86.71 $\pm$ 0.26 | **87.55** $\pm$ 0.02 | 89.20 | 88.50 |
| SUN | 60.55 | 60.39 $\pm$ 0.12 | 63.51 $\pm$ 0.03 | 61.66 $\pm$ 0.17 | **64.95** $\pm$ 0.10 | 75.00 | 64.00 |
| SVHN | 15.30 | 91.61 $\pm$ 0.12 | 86.69 $\pm$ 0.09 | **92.94** $\pm$ 0.18 | 91.81 $\pm$ 0.18 | 65.40 | 95.70 |
| UCF | 61.46 | 65.68 $\pm$ 0.28 | 67.69 $\pm$ 0.52 | 70.88 $\pm$ 0.11 | **76.11** $\pm$ 0.38 | 83.30 | 80.90 |
| Average | 53.12 | 78.39 | 78.86 | 81.54 | **83.18** | 82.59 | 87.37 |

consistent improvements across diverse datasets highlight the enhanced generalization capability of ACAVP. Notably, ACAVP shows remarkable improvement on the Flowers dataset, with absolute gains of 6.67 percentage points over AutoVP and 21.53 percentage points over the VP. This substantial improvement suggests that our affine transformation and color transformation components are particularly effective for datasets with fine-grained visual features, such as the intricate textures and shapes found in flower images. The ability to capture such nuanced visual characteristics demonstrates the enhanced expressive power of ACAVP. Interestingly, ACAVP surpasses the average accuracy of Linear Probing (LP), which modifies the model parameters. VP methods, including ours, demonstrate particularly strong performance on datasets that differ significantly from the model's pretraining distribution. For instance, on specialized datasets such as GTSRB (traffic signs) and SVHN (street view house numbers), VP approaches substantially outperform LP. This pattern aligns with previous findings that VP is especially effective for adapting to domains that deviate from the source distribution used during pretraining [Bahng et al., 2022]. In summary, our experimental results demonstrate that incorporating affine transformation and color transformation alongside overfitting mitigation techniques effectively addresses the limitations of conventional VP approaches. These results validate our hypothesis that diversifying the transformation space and mitigating overfitting are crucial factors for enhancing the effectiveness of VP techniques.

We further evaluated ACAVP using ResNet50 [He et al., 2016] pre-trained on the ImageNet-1k dataset [Russakovsky et al., 2015] across the same 12 image classification datasets. This evaluation aims to demonstrate the generalizability of our approach across different model architectures and pretraining strategies. Table 3 shows the results. Similar to our findings with the CLIP model, ACAVP outperforms baseline approaches on many datasets and achieves the highest average accuracy among all VP methods. Specifically, without label mapping optimization (left columns), ACAVP achieves the highest average accuracy, significantly outperforming the second-best method by 15.01 percentage points and 21.50 percentage points on the Flowers and GTSRB datasets, respectively. Similarly, with label mapping (denoted by $^\dagger$ in the right columns), ACAVP achieves the highest average accuracy. While all methods benefit from label mapping, our approach maintains its superior performance, suggesting that the benefits of our enhanced transformation space are complementary to those provided by label mapping. The consistent relative performance across both settings indicates that ACAVP's improvements stem from fundamental enhancements to the VP framework rather than interactions with specific implementation details. These results demonstrate that ACAVP generalizes effectively across different model architectures and pretraining strategies. The consistent performance improvements observed with both CLIP and ResNet50 backbones suggest that the limitations of conventional VP approaches are fundamental rather than model-specific and that our enhancements to the transformation space and learning methodology principledly address these limitations.

Table 3: Classification accuracy comparison on 12 image classification datasets using ResNet50 pre-trained on ImageNet-1k. $\dagger$ indicates results with label mapping proposed by Wu et al. [2024].

| Method | VP | EVP | AutoVP | ACAVP | VP$^\dagger$ | EVP$^\dagger$ | AutoVP$^\dagger$ | ACAVP$^\dagger$ |
|---|---|---|---|---|---|---|---|---|
| CIFAR10 | 55.09 $\pm$ 0.25 | 54.28 $\pm$ 0.20 | 45.52 $\pm$ 0.34 | 53.48 $\pm$ 0.61 | 64.09 $\pm$ 0.08 | 67.93 $\pm$ 0.26 | 62.77 $\pm$ 0.35 | 72.92 $\pm$ 0.91 |
| CIFAR100 | 10.95 $\pm$ 0.15 | 7.40 $\pm$ 0.26 | 6.80 $\pm$ 0.36 | 14.84 $\pm$ 0.08 | 27.69 $\pm$ 0.11 | 32.30 $\pm$ 0.29 | 29.44 $\pm$ 0.39 | 33.71 $\pm$ 0.56 |
| CLEVR | 42.19 $\pm$ 0.19 | 40.62 $\pm$ 0.47 | 28.90 $\pm$ 0.49 | 32.23 $\pm$ 0.06 | 40.23 $\pm$ 0.34 | 37.88 $\pm$ 0.24 | 29.16 $\pm$ 0.41 | 38.95 $\pm$ 0.49 |
| DTD | 13.46 $\pm$ 0.36 | 15.98 $\pm$ 0.61 | 12.59 $\pm$ 0.24 | 9.34 $\pm$ 2.85 | 30.65 $\pm$ 0.15 | 35.62 $\pm$ 0.52 | 30.04 $\pm$ 0.26 | 34.16 $\pm$ 0.77 |
| EuroSAT | 79.06 $\pm$ 0.08 | 78.44 $\pm$ 0.11 | 68.79 $\pm$ 0.64 | 76.70 $\pm$ 1.16 | 82.18 $\pm$ 0.20 | 79.94 $\pm$ 0.14 | 71.82 $\pm$ 0.99 | 83.04 $\pm$ 0.23 |
| Flowers | 8.40 $\pm$ 0.35 | 10.37 $\pm$ 0.43 | 10.22 $\pm$ 0.20 | 25.38 $\pm$ 0.63 | 12.13 $\pm$ 0.06 | 16.59 $\pm$ 0.65 | 14.60 $\pm$ 0.24 | 19.77 $\pm$ 0.62 |
| Food | 4.22 $\pm$ 0.17 | 3.24 $\pm$ 0.01 | 2.82 $\pm$ 0.04 | 6.69 $\pm$ 0.05 | 7.66 $\pm$ 0.09 | 7.14 $\pm$ 0.19 | 7.18 $\pm$ 0.09 | 8.55 $\pm$ 0.55 |
| GTSRB | 38.87 $\pm$ 0.43 | 30.26 $\pm$ 0.60 | 28.17 $\pm$ 0.64 | 60.37 $\pm$ 0.73 | 45.89 $\pm$ 0.10 | 34.93 $\pm$ 0.39 | 37.66 $\pm$ 0.52 | 45.77 $\pm$ 0.05 |
| Pets | 5.76 $\pm$ 0.11 | 7.47 $\pm$ 0.18 | 6.92 $\pm$ 0.14 | 12.23 $\pm$ 0.25 | 64.80 $\pm$ 0.32 | 68.66 $\pm$ 0.08 | 66.68 $\pm$ 0.07 | 68.23 $\pm$ 0.27 |
| SUN | 0.87 $\pm$ 0.02 | 1.10 $\pm$ 0.08 | 0.60 $\pm$ 0.07 | 1.50 $\pm$ 0.03 | 7.93 $\pm$ 0.03 | 8.73 $\pm$ 0.16 | 7.74 $\pm$ 0.08 | 7.33 $\pm$ 0.16 |
| SVHN | 66.54 $\pm$ 0.32 | 48.21 $\pm$ 0.30 | 46.60 $\pm$ 1.51 | 62.21 $\pm$ 0.48 | 67.84 $\pm$ 0.21 | 49.17 $\pm$ 0.47 | 45.33 $\pm$ 1.24 | 68.82 $\pm$ 0.44 |
| UCF | 8.19 $\pm$ 0.33 | 7.28 $\pm$ 0.66 | 6.53 $\pm$ 0.10 | 13.11 $\pm$ 0.19 | 24.50 $\pm$ 0.21 | 27.56 $\pm$ 0.07 | 26.14 $\pm$ 0.23 | 28.14 $\pm$ 0.75 |
| Average | 27.80 | 25.39 | 22.04 | 30.67 | 39.63 | 38.87 | 35.71 | 42.45 |

Table 4: Classification accuracy on the corrupted datasets using CLIP ViT-B/32. Classification accuracy means the average accuracy at 15 different types and 5 levels of severity.

| Method | CIFAR10-C | CIFAR100-C |
|---|---|---|
| ZS | 70.9 | 42.59 |
| VP | 76.65 $\pm$ 0.04 | 50.52 $\pm$ 0.12 |
| EVP | 80.97 $\pm$ 0.14 | 55.17 $\pm$ 0.16 |
| AutoVP | 79.72 $\pm$ 0.28 | 56.34 $\pm$ 0.15 |
| ACAVP | 83.98 $\pm$ 0.20 | 58.68 $\pm$ 0.29 |

Table 5: An ablation study shows the contribution of different transformation components. Affine refers to affine transformation, Additive to additive transformation, and Color to color transformation. All methods use TrivialAugment for fair comparison.

| Transformation | CIFAR10 | Flowers |
|---|---|---|
| – | 88.93 | 61.63 |
| Affine | 90.17 $\pm$ 0.01 | 57.42 $\pm$ 0.03 |
| Color | 92.14 $\pm$ 0.06 | 62.92 $\pm$ 0.04 |
| Affine + Color | 92.46 $\pm$ 0.04 | 60.58 $\pm$ 0.08 |
| Resize + Additive | 96.00 $\pm$ 0.05 | 79.24 $\pm$ 0.18 |
| Resize + Additive + Color | 96.05 $\pm$ 0.04 | 80.74 $\pm$ 0.28 |
| Affine + Additive | 96.31 $\pm$ 0.02 | 88.70 $\pm$ 0.18 |
| Affine + Color + Additive | 96.27 $\pm$ 0.07 | 89.36 $\pm$ 0.19 |

## 4.2 Robustness to Distribution Shift

We evaluated the robustness of ACAVP to distribution shifts by comparing its performance against baseline approaches on artificially corrupted datasets. Distribution robustness is a critical aspect of adaptation methods in real-world scenarios where test data often differs from training data due to various forms of noise and corruption. For this evaluation, we tested various VP approaches on the CIFAR10-C and CIFAR100-C datasets, which apply 15 common types of image corruptions (including noise, blur, weather, and digital distortions) at different severity levels to the original CIFAR10 and CIFAR100 test sets, respectively [Hendrycks and Dietterich, 2019]. These standardized corruption benchmarks provide a systematic way to assess model robustness under various distribution shifts. The results are shown in Table 4. ACAVP consistently outperforms all baseline approaches by significant margins on both corrupted datasets. Notably, the performance improvement of ACAVP is more pronounced on out-of-distribution data compared to in-distribution performance (refer to Table 2). For example, on the CIFAR10-C dataset, ACAVP achieves a 7.33 percentage points improvement over VP, which is higher than the 2.62 percentage points improvement observed on the original CIFAR10 dataset. This enhanced robustness to distribution shifts suggests that the diverse transformation space introduced by ACAVP, combined with overfitting mitigation techniques, enables learning more generalizable transformations. These results highlight an important practical advantage of ACAVP: improved robustness to real-world distribution shifts, which is often crucial for the deployment of adaptation methods in dynamic and unpredictable environments.

## 4.3 Ablation Study

We conducted an ablation study to analyze the contribution of individual components in ACAVP on the CIFAR10 and Flowers datasets. This analysis helps identify which transformations are most effective and how they interact with each other. Using ZS (no transformation) and EVP (which

Table 6: Comparison of overfitting mitigation techniques on image-sized VP with CLIP ViT-B/32. Values in parentheses show accuracy gains from using overfitting mitigation techniques.

| Method | CIFAR10 | Flowers |
|---|---|---|
| Dropout | 88.50 $_{\pm 0.18}$ (-1.02) | 53.57 $_{\pm 0.41}$ (-0.28) |
| MSE Loss | 90.26 $_{\pm 0.02}$ (+0.74) | 53.74 $_{\pm 0.34}$ (-0.11) |
| Weight decay | 91.25 $_{\pm 0.10}$ (+1.73) | 55.37 $_{\pm 0.58}$ (+1.52) |
| TrivialAugment | 91.56 $_{\pm 0.06}$ (+2.04) | 70.11 $_{\pm 0.61}$ (+16.26) |
| IPMix | 90.85 $_{\pm 0.07}$ (+1.33) | 66.03 $_{\pm 0.80}$ (+12.18) |
| RandAugment | 91.52 $_{\pm 0.05}$ (+2.00) | 72.88 $_{\pm 0.70}$ (+19.03) |

applies an additive transformation after resizing) as the baselines, we evaluated six variants: affine transformation alone (Affine), color transformation alone (Color), affine and color transformations (Affine + Color), EVP with color transformation (Resize + Additive + Color), EVP with affine transformation replacing the resize operation (Affine + Additive), and ACAVP that combines affine transformation, color transformation, and additive transformation (Affine + Color + Additive). To ensure a fair comparison, we incorporated TrivialAugment in the training process for all methods, including the EVP baseline. The results are shown in Table 5. Individual transformations show varying effectiveness across datasets: the affine transformation improves performance on the CIFAR10 dataset but shows reduced performance on the Flowers dataset compared to ZS, while the color transformation consistently outperforms ZS accuracy on both datasets. Combining affine and color transformations yields further improvements. On the Flowers dataset, which contains rich textural and color information, the performance gains are particularly illuminating. The variant with color transformation (Resize + Additive + Color) outperforms EVP by 1.50 percentage points, while the variant with affine transformation (Affine + Additive) exceeds EVP by 9.46 percentage points. These results demonstrate the significant contributions of both color transformation and affine transformation to accuracy, with affine transformation having a more substantial impact on the Flowers dataset. Notably, our full method, which incorporates both affine and color transformations, achieves the highest accuracy, surpassing EVP by 10.12 percentage points on the Flowers dataset. This suggests that these transformations are complementary rather than competing, and their combination leads to more effective adaptation. The affine transformation likely helps in focusing on the most discriminative regions of flowers, while the color transformation enhances the representation of subtle color variations that are critical for flower classification. On the CIFAR10 dataset, all variants show improvements over ZS, though the differences are less pronounced than on the Flowers dataset. This smaller gap is expected since the CIFAR10 features less complex visual patterns and color variations than the Flowers. Nevertheless, the improvements across both datasets confirm that each component contributes positively to the overall performance of our method. This ablation study validates our design choices and demonstrates that both the affine and color transformation components contribute meaningfully to improving VP performance.

### 4.4 Effectiveness of Overfitting Mitigation Strategies

We demonstrate experimentally that mitigating overfitting plays a crucial role in VP training. Our preliminary experiments showed that VP approaches, particularly those using image-sized noise additions, exhibit strong overfitting tendencies that limit performance. Here, we investigate various strategies to address this limitation. We first experimented applying various overfitting mitigation techniques to the image-sized VP approach, which showed overfitting tendencies in our preliminary studies. The techniques we evaluated include: dropout layer after the feature extraction layer, MSE loss minimization between the original and prompted images, weight decay for parameter regularization, and three data augmentation methods, TrivialAugment, RandAugment [Cubuk et al., 2020], and IPMix [Huang et al., 2023], which increase input diversity. The results are shown in Table 6. Dropout and MSE show no significant improvement over the baseline. Weight decay demonstrates a modest accuracy improvement. Most notably, data augmentation techniques yield substantial performance gains, with TrivialAugment and RandAugment showing powerful improvements on both datasets. These results strongly suggest that overfitting mitigation is critical for VP training, with data augmentations being especially effective.

Next, we investigated whether TrivialAugment, the effective and computationally efficient regularization technique from our first experiment, could improve the performance of existing VP baselines.

Table 7: Effect of TrivialAugment on existing VP methods with CLIP ViT-B/32. Values in parentheses show accuracy gains from adding TrivialAugment.

| Method | CIFAR10 | Flowers |
|--------|---------|---------|
| VP | 94.97 ± 0.06 (+1.32) | 79.84 ± 0.07 (+12.01) |
| EVP | 96.00 ± 0.05 (+0.13) | 79.24 ± 0.18 (+1.48) |
| AutoVP | 96.16 ± 0.02 (+0.77) | 85.83 ± 0.58 (+3.14) |

The results, shown in Table 7, demonstrate consistent accuracy improvements across all VP baselines when TrivialAugment is applied. The improvements are particularly pronounced for the standard VP approach on the Flowers dataset, with a substantial 12.01 percentage point accuracy gain. Interestingly, with TrivialAugment, the standard VP achieves accuracy comparable to EVP, despite EVP's additional resize operation. This suggests that the resize operation in EVP may have been functioning primarily as an implicit regularizer by preserving more of the original image information and limiting overfitting. However, this advantage diminishes with explicit input diversity through TrivialAugment, indicating that VP and EVP may have similar expressive power when properly regularized. These findings strongly support our approach of diversifying the transformation space while addressing overfitting. They suggest that the performance limitations of conventional VP methods stem not only from restricted transformation operations but also from their susceptibility to overfitting. By combining a more expressive transformation space with effective regularization techniques, ACAVP addresses both limitations simultaneously, leading to substantial performance improvements across diverse datasets.

## 4.5 Computational Efficiency Analysis

Table 8: Inference time comparison across different methods using H100 GPU with batch size 500. Relative Time is normalized to the CLIP ViT-B/32's inference time.

| Method | Time (s) | Relative Time |
|--------|----------|---------------|
| VP | $2.88 \times 10^{-5}$ | 0.00 |
| EVP | $1.38 \times 10^{-4}$ | 0.00 |
| AutoVP | $1.77 \times 10^{-2}$ | 0.60 |
| ACAVP | $1.63 \times 10^{-3}$ | 0.06 |
| Coordinator | $2.01 \times 10^{-1}$ | 6.85 |
| Model Inference | $2.93 \times 10^{-2}$ | 1.00 |

A key advantage of VP is their minimal computational overhead during inference, as they modify only the input images without altering the recognition model. While ACAVP enhances accuracy by incorporating affine and color transformations, these additional operations naturally introduce some computational costs. In this section, we measured the inference time of different VP methods. For comparison purposes, we also measured the inference time of the recognition model itself (CLIP ViT-B/32) and a neural network-based VP approach called Coordinator [Oh et al., 2023], which uses an encoder-decoder architecture to generate visual prompts. The results are presented in Table 8. As expected, ACAVP requires more computation time than VP and EVP. However, when compared to the model's inference time, ACAVP adds only 6% of the model's computational cost, which is negligible in practical applications. Notably, the Coordinator has a longer inference time than the recognition model inference. This substantial overhead undermines one of the key benefits of VP approaches: their minimal impact on inference efficiency. Our approach strikes an optimal balance between computational efficiency and transformation expressivity. Using linear transformations (affine and color) rather than complex neural networks, we achieve significant improvements in accuracy with only a modest increase in computational cost. This makes ACAVP particularly attractive for real-world applications where performance and efficiency are important considerations. These results highlight an important design principle for VP approaches: while enhancing the transformation space is crucial for improving accuracy, the choice of transformation operations should consider their computational implications. Linear transformations like those used in ACAVP offer a favorable trade-off: They provide substantial expressive power with minimal computational overhead.

## 5 Conclusion

In this paper, we addressed the limitations of conventional visual prompting (VP) techniques by introducing ACAVP, a novel framework that enhances VP's expressive power and mitigates over-

fitting issues. Our approach builds upon the existing VP paradigm by incorporating two additional transformation operations: affine transformation, which creates task-specific prompt regions while preserving original image information, and color transformation, which emphasizes task-relevant features through brightness and contrast adjustments. Furthermore, we demonstrated that data augmentation techniques, particularly TrivialAugment, significantly improve generalization performance for VP methods by mitigating overfitting tendencies. Our extensive experimental evaluation across 12 diverse image classification datasets using both CLIP ViT and ResNet50 architectures validates the effectiveness of ACAVP. ACAVP consistently outperforms existing VP methods and achieves higher average accuracy than linear probing, which modifies model parameters. Notably, our method exhibits superior robustness to distribution shifts, as demonstrated by its performance on corrupted datasets. These improvements are achieved while maintaining VP's key advantage of minimal computational overhead during inference, with ACAVP adding only $6\%$ to the model's inference time. The success of ACAVP highlights the importance of transformation expressivity and regularization in VP training.

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

# A Related Work

## A.1 Applicability of VP.

VP has been successfully applied to black-box scenarios where model internals are inaccessible. Tsai et al. [2020] proposed black-box adversarial reprogramming (BAR), which leverages zeroth-order optimization, using only numerical evaluations of the loss function instead of gradients. Oh et al. [2023] further advanced this direction with BlackVIP, enabling adaptation without knowledge of the recognition model's architecture or parameters. VP has shown promise in adapting models at test time to handle distribution shifts. Gan et al. [2023] applied VP to test-time adaptation, improving inference accuracy on out-of-distribution data by updating prompts during test time without labels. Tsai et al. [2023] introduced convolutional visual prompt (CVP), which are pre-trained through self-supervised learning and adapted at test time to enhance out-of-distribution performance.

## A.2 Visual Prompt Tuning.

Visual prompt tuning (VPT) [Jia et al., 2022, Han et al., 2024, 2023, Zeng et al., 2024] is conceptually related to visual prompting. VPT adapts to downstream tasks by optimizing input embeddings as well as the classification head of the model. In contrast, VP methods attach learnable parameters directly to the input image, and typically adjust the classification head using heuristic methods [Wu et al., 2024, Chen et al., 2023, Cai et al., 2024a], rather than end-to-end optimization.

Importantly, VP methods are often designed for settings where the architecture of the recognition model is unknown or the gradients are inaccessible, as in black-box scenarios [Tsai et al., 2020, Oh et al., 2023].

# B Theoretical Analysis

In this section, we provide a theoretical analysis of ACAVP within the framework of probably approximately correct (PAC) learning [Kearns and Vazirani, 1994]. We establish that our proposed method has a smaller approximation error compared to existing visual prompting approaches, which explains its superior empirical performance.

Our analysis draws inspiration from the theoretical framework established by Cai et al. [2024b] for analyzing Sample-specific Masks for Visual Reprogramming (SMM). Cai et al. [2024b] demonstrated that using sample-specific masks reduces the approximation error compared to shared masks in visual reprogramming. We extend this insight to analyze the relationship between different visual prompting methods.

According to Cai et al. [2024b], the approximation error measures how close the best achievable error by a hypothesis space is to the theoretical minimum error on a distribution. When hypothesis spaces exhibit a subset relation, the larger hypothesis space has a smaller approximation error.

**Theorem B.1** (Cai et al. [2024b]). *Given an input space $X$, a discrete label space $Y$, and a distribution $D$ over $X \times Y$, if there are two hypothesis spaces $F_1 \subseteq \{f : X \to Y\}$ and $F_2 \subseteq \{f : X \to Y\}$ satisfying $F_1 \subseteq F_2$, then we have*

$$Err_D^{apx}(F_1) \geq Err_D^{apx}(F_2). \tag{3}$$

## B.1 Hypothesis Spaces of Visual Prompting Methods

We now formalize the hypothesis spaces of the three visual prompting methods under consideration. For a fixed pre-trained model $f_P$ and fixed output mapping function $f_{\text{out}}$ (where $f'_P = f_{\text{out}} \circ f_P$), we define:

**EVP Hypothesis Space.** Enhanced Visual Prompting (EVP) resizes the input image by a fixed factor $s$ and adds a learned pattern $\delta$. Since the resize factor is predetermined, a fixed mask $M$ can be prepared in advance. The hypothesis space of EVP is defined as:

$$F^{\text{evp}}(f'_P) = \{f|f(x) = f'_P(r_s(x) + M \odot \delta), \forall x \in X\}, \tag{4}$$

where $r_s(\cdot)$ is a resizing operation with fixed scale factor $s$, $M$ is the predetermined mask, and $\delta$ is the learnable pattern.

**AutoVP Hypothesis Space.** AutoVP extends EVP by making the resizing factor learnable. Since the image size changes during training, the mask is dynamically created according to the resize parameter. Its hypothesis space is defined as:

$$F^{\text{autovp}}(f'_P) = \{f | f(x) = f'_P(r_{\hat{s}}(x) + M \odot \delta), \forall x \in X\}, \tag{5}$$

where $\hat{s}$ is a learnable scaling parameter, $M$ is the mask dynamically generated based on $\hat{s}$, and $\delta$ is the learnable pattern.

**ACAVP Hypothesis Space.** Our proposed ACAVP incorporates affine and color transformation. Its hypothesis space is defined as:

$$F^{\text{acavp}}(f'_P) = \{f | f(x) = f'_P(\text{Affine}(x, A) \odot \hat{\sigma} + M \odot \delta), \forall x \in X\}, \tag{6}$$

where $\text{Affine}(\cdot, A)$ represents the affine transformation with learnable affine prompt $A \in \mathbb{R}^{3 \times 3}$, $\hat{\sigma} \in \mathbb{R}^{3 \times H \times W}$ is the learnable color prompt, and $\delta$ is the learnable additive prompt.

## B.2 Relationship Between Hypothesis Spaces

Following the approach in Cai et al. [2024b], we can establish the following relationship between these hypothesis spaces:

**Proposition B.2.** *For any fixed pre-trained model $f_P$ and fixed output mapping function $f_{out}$, we have*

$$F^{evp}(f'_P) \subseteq F^{autovp}(f'_P) \subseteq F^{acavp}(f'_P). \tag{7}$$

*Proof.* First, we show that $F^{\text{evp}}(f'_P) \subseteq F^{\text{autovp}}(f'_P)$. For any function $f \in F^{\text{evp}}(f'_P)$, we have

$$f(x) = f'_P(r_s(x) + M \odot \delta) \tag{8}$$

for some fixed scale factor $s$ and learnable pattern $\delta$. We can find a corresponding function in $F^{\text{autovp}}(f'_P)$ by setting the learnable parameter $\hat{s} = s$, resulting in

$$f(x) = f'_P(r_{\hat{s}}(x) + M \odot \delta) \tag{9}$$

with the same $\delta$. Therefore, $f \in F^{\text{autovp}}(f'_P)$, which means $F^{\text{evp}}(f'_P) \subseteq F^{\text{autovp}}(f'_P)$.

Next, we show that $F^{\text{autovp}}(f'_P) \subseteq F^{\text{acavp}}(f'_P)$. For any function $f \in F^{\text{autovp}}(f'_P)$, we have

$$f(x) = f'_P(r_{\hat{s}}(x) + M \odot \delta) \tag{10}$$

for some learnable parameter $\hat{s}$ and pattern $\delta$. We can construct a corresponding function in $F^{\text{acavp}}(f'_P)$ by:

- Setting the affine transformation matrix $A = \begin{bmatrix} \hat{s} & 0 & 0 \\ 0 & \hat{s} & 0 \\ 0 & 0 & 1 \end{bmatrix}$, which performs only scaling by $\hat{s}$

- Setting the color transformation prompt $\hat{\sigma} = \mathbf{1}$ (all ones)

- Using the same additive prompt $\delta$

With these settings, we have:

$$\text{Affine}(x, A) = r_{\hat{s}}(x) \tag{11}$$
$$\text{Affine}(x, A) \odot \hat{\sigma} = r_{\hat{s}}(x) \odot \mathbf{1} = r_{\hat{s}}(x) \tag{12}$$

Therefore,

$$f(x) = f'_P(\text{Affine}(x, A) \odot \hat{\sigma} + M \odot \delta) = f'_P(r_{\hat{s}}(x) + M \odot \delta) \tag{13}$$

This shows that $f \in F^{\text{acavp}}(f'_P)$, which means $F^{\text{autovp}}(f'_P) \subseteq F^{\text{acavp}}(f'_P)$. $\square$

Based on Theorem B.1 and Proposition B.2, we can establish our main theoretical result:

**Proposition B.3.** *For any fixed pre-trained model $f_P$, fixed output mapping function $f_{out}$, and distribution $D_T$ of the target task, we have*

$$Err_{D_T}^{apx}(F^{evp}(f_P')) \geq Err_{D_T}^{apx}(F^{autovp}(f_P')) \geq Err_{D_T}^{apx}(F^{acavp}(f_P')). \tag{14}$$

This theoretical result, following the framework proposed by Cai et al. [2024b], explains why ACAVP achieves superior performance compared to EVP and AutoVP. By expanding the hypothesis space through the introduction of affine and color transformation, ACAVP can more effectively approximate the optimal function for the target task.

Similar to the observation in Cai et al. [2024b] regarding estimation error, we note that while a larger hypothesis space typically reduces approximation error, it may increase estimation error due to the added complexity. However, our experimental results show that this potential increase in estimation error is well-managed through our training approach, particularly through the use of TrivialAugment for overfitting mitigation.

## C  Implementation Details

In practice, our ACAVP method applies the transformations in the following order: first, the affine transformation is applied to the input image; then, the color transformation is applied through element-wise multiplication; finally, the additive noise prompt is applied in the designated regions specified by the mask $M$.

For initialization, we set the affine transformation parameters to perform a resizing operation, inspired by the success of a previous study [Wu et al., 2024]. Specifically, we initialize the parameters to resize a 224×224 image to 164×164, setting $\hat{s} = 0.73$ (which corresponds to $s = \text{sigmoid}^{-1}(0.73)$) and $t = \theta = sh = 0$. This initialization leverages the effectiveness of resizing operations demonstrated in previous research while providing a good starting point for optimization. The color transformation prompt is initialized to $\hat{\sigma} = 1$, corresponding to no color transformation of the original image. The additive prompt $\delta$ is initialized to zero, following the initialization strategy used in [Wu et al., 2024]. This initialization strategy ensures that the training begins with a controlled transformation (modest resizing) while maintaining the essential visual information, allowing the model to gradually learn more complex and task-specific transformations.

During the training process, all three components of ACAVP—affine, color, and additive transformations—are jointly optimized using stochastic gradient descent to minimize the task-specific loss function. The TrivialAugment data augmentation is applied to each mini-batch before the prompting transformations, providing the necessary input diversity to prevent overfitting.

## D  Experimental Setup

### D.1  Datasets

We evaluated our method on 12 downstream classification tasks: CIFAR100, CIFAR10 [Krizhevsky et al., 2009], Flowers102 [Nilsback and Zisserman, 2008], Food101 [Bossard et al., 2014], EuroSAT [Helber et al., 2019], SUN397 [Xiao et al., 2010], DTD [Cimpoi et al., 2014], UCF101 [Soomro et al., 2012], SVHN [Netzer et al., 2011], OxfordPets [Parkhi et al., 2012], GTSRB [Houben et al., 2013], and CLEVR [Johnson et al., 2017]. These datasets encompass diverse visual domains, including natural scenes, human actions, textures, and fine-grained object details, providing a comprehensive evaluation of our method's generalization capability.

For dataset splits, we follow the CoOp protocol [Zhou et al., 2022b,a]. For datasets with predefined validation sets, we use them directly. For others, we randomly allocate $10\%$ of the training data for validation and use the remaining $90\%$ for training. We select the checkpoint with the highest validation accuracy for final evaluation. Detailed dataset information is provided in Table 9.

### D.2  Model Architectures

We evaluate ACAVP on two representative architectures: CLIP ViT-B/32 [Radford et al., 2021] and ImageNet-1k [Russakovsky et al., 2015] pre-trained ResNet50 [He et al., 2016]. For CLIP

Table 9: Detailed dataset information. $^\dagger$ indicates that data splits provided by CoOp were used.

| Dataset | Train Size | Validation Size | Test Size | Classes | Image Size |
|---|---|---|---|---|---|
| CIFAR100 | 45,000 | 5,000 | 10,000 | 100 | 32×32 |
| CIFAR10 | 45,000 | 5,000 | 10,000 | 10 | 32×32 |
| Flowers102$^\dagger$ | 4,093 | 1,633 | 2,463 | 102 | 128×128 |
| Food101$^\dagger$ | 50,500 | 20,200 | 30,300 | 101 | 128×128 |
| EuroSAT$^\dagger$ | 13,500 | 5,400 | 8,100 | 10 | 128×128 |
| SUN397$^\dagger$ | 15,880 | 3,970 | 19,850 | 397 | 128×128 |
| UCF101$^\dagger$ | 7,639 | 1,898 | 3,783 | 101 | 128×128 |
| SVHN | 65,931 | 7,326 | 26,032 | 10 | 32×32 |
| OxfordPets$^\dagger$ | 2,944 | 736 | 3,669 | 37 | 128×128 |
| DTD$^\dagger$ | 2,820 | 1,128 | 1,692 | 47 | 128×128 |
| GTSRB | 35,288 | 3,921 | 12,630 | 43 | 32×32 |
| CLEVR | 63,000 | 7,000 | 15,000 | 8 | 480×320 |

experiments, we use the prompt template "This is a photo of a [LABEL]" for all datasets except CLEVR, where we use "This is a photo of [LABEL] objects" to better match its task-specific nature. For the ImageNet pre-trained model, we experiment with two label mapping strategies: random label mapping, which randomly assigns ImageNet classes to target dataset classes, and the label mapping method proposed by [Wu et al., 2024].

### D.3 Baselines

We compare ACAVP against several baseline methods. Zero-shot inference (ZS) using CLIP serves as our basic baseline without any adaptation. We also compare against standard visual prompting (VP) [Bahng et al., 2022] that adds learnable prompts around the input image, Enhanced VP (EVP) [Wu et al., 2024] that applies fixed resizing before adding prompts, and AutoVP [Tsao et al., 2024] that learns the resize ratio parameter along with the prompt. These baselines represent the current state-of-the-art visual prompting techniques, comprehensively evaluating our method's advantages.

### D.4 Training details

In our training setup, we used an initial learning rate of 40 with a Cosine Annealing Learning Rate Scheduler to gradually decrease the learning rate throughout training. The total number of epochs was set to 1000 for all experiments. We employed SGD as our optimizer with a momentum of 0.9. For most datasets, we used a batch size of 256, with the exception of DTD dataset where we used a batch size of 64 due to its smaller size. For input preprocessing, we first resized all training images to 256×256 pixels, then performed center cropping to 224×224 pixels, followed by normalization. For EVP training, we additionally applied horizontal flip as a data augmentation technique and implemented gradient normalization to stabilize training. When training AutoVP with CLIP models, we used gradient clipping via torch.nn.utils.clip_grad_value_ to limit the maximum gradient value to 0.001. Similarly, for ACAVP training, we applied gradient clipping with a maximum value of 0.001, and when using CLIP models, we also employed gradient normalization for improved stability.

## E Additional Experiments

### E.1 Impact of Data Augmentation on ACAVP

We analyze the effectiveness of different data augmentation techniques when applied to ACAVP training. The results of our comparison are presented in Table 10. As shown in the table, all augmentation techniques improved the performance of ACAVP compared to training without augmentation.

### E.2 Visualization Analysis

We show visualizations of the transformed images generated by different VP methods and the Grad-CAM [Selvaraju et al., 2020] activation maps in Figure 2. For the CLIP ViT model, we observe that

Table 10: Comparison of data augmentation techniques on ACAVP with CLIP ViT-B/32.

| Data Augmentation | CIFAR10 | Flowers |
|---|---|---|
| – | 95.74 $_{\pm 0.03}$ | 87.55 $_{\pm 0.64}$ |
| IPIMix | 95.96 $_{\pm 0.03}$ | 90.65 $_{\pm 0.27}$ |
| RandAugment | 96.01 $_{\pm 0.10}$ | 88.24 $_{\pm 0.18}$ |
| TrivialAugment | 96.27 $_{\pm 0.07}$ | 89.36 $_{\pm 0.19}$ |

| VP | EVP | AutoVP | ACAVP |
|---|---|---|---|

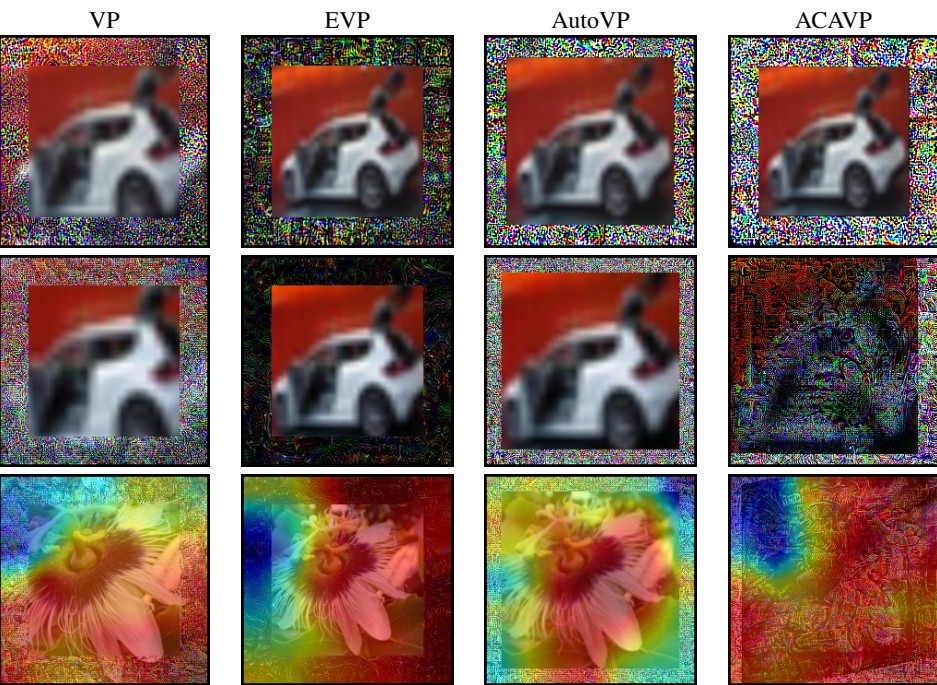

Figure 2: Visualization analysis of different VP methods. The first row shows transformed images on the CIFAR10 dataset using CLIP ViT-B/32. The second row presents transformed images on the CIFAR10 dataset using ResNet50. The third row displays Grad-CAM activation maps on the Flowers dataset using ResNet50.

ACAVP applies a minimal affine transformation, appearing similar to the initial resize operation. This conservative transformation is logical considering that CLIP ViT already achieves high zero-shot classification accuracy on the CIFAR10 dataset (88.93% as shown in Table 2). When a model has strong prior knowledge about the target domain, extensive transformations may be unnecessary or even detrimental, and our method adaptively learns this characteristic. In contrast, when applied to ResNet50 on the CIFAR10 dataset, ACAVP performs more substantial transformations compared to other VP approaches. This is consistent with ResNet50's lower zero-shot classification capabilities on this dataset, necessitating more significant input modifications to improve accuracy. The ability of ACAVP to adapt its transformation magnitude based on the backbone model inherent capabilities demonstrates its flexibility across different model architectures. The capacity of ACAVP for more expressive transformations stems from the inclusion of both affine transformation and color transformation, which provide a richer transformation space compared to conventional VP approaches that rely solely on additive noise. This enhanced flexibility allows our method to apply appropriate transformations based on the specific requirements of the dataset and backbone model combination. The Grad-CAM visualization results reveal another important advantage of our approach. ACAVP produces higher activation values across broader regions of the image compared to other VP methods, indicating more comprehensive utilization of the image information. This suggests that our approach enables the model to consider a wider range of visual features during classification, rather than focusing on limited regions as observed with other VP methods.

Table 11: Classification accuracy comparison on the CIFAR10 and Flowers datasets using various models. The parentheses indicate model architecture specifications. [†] indicates results with label mapping proposed by Wu et al. [2024].

| Model | ResNet101[†] | | ViT-L/16[†] | | DINOv2(ViT-S/14)[†] | | SigLIP(ViT-SO400M/14) | | Average |
|---|---|---|---|---|---|---|---|---|---|
| Method | CIFAR10 | Flowers | CIFAR10 | Flowers | CIFAR10 | Flowers | CIFAR10 | Flowers | |
| ZS | 57.32 | 5.16 | 69.49 | 4.95 | 72.35 | 8.89 | 95.30 | 88.96 | 50.30 |
| VP | 67.44 | 10.72 | 90.69 | 13.60 | 81.63 | 24.69 | 96.41 | 97.00 | 60.27 |
| EVP | 72.86 | 13.52 | 94.96 | 27.49 | 91.63 | 10.23 | 98.76 | 98.66 | 63.51 |
| AutoVP | 68.82 | 13.20 | 94.75 | 25.74 | **94.25** | **43.85** | 98.72 | 97.48 | 67.10 |
| ACAVP | **75.99** | **17.66** | **95.38** | **39.71** | 93.62 | 38.49 | **98.83** | **98.74** | **69.80** |

## E.3 Hyperparameters sensitivity

In this section, we investigate the sensitivity of ACAVP to its key hyperparameters. The ACAVP introduces several hyperparameters that control the range of transformations applied to input images. Specifically, we analyze the impact of hyperparameters $R_\theta$, $R_{sh}$, and $R_t$ that restrict the range of affine transformation, and $R_\sigma$ that limits the range of color transformation.

We conducted experiments using the CLIP ViT-B/32 model on the CIFAR10 and Flowers datasets. For the affine transformation parameters, we varied the scaling factor $X$ applied to the default values ($R_t = 0.05$, $R_\theta = R_{sh} = 0.1$). For the color transformation parameter $R_\sigma$, we tested values ranging from 2.5 to 20.0, with the default value being 6.0. All other training settings remained consistent with our main experimental setup.

The results of our hyperparameter sensitivity analysis are presented in Figure 3. As shown in the figure, the performance of ACAVP exhibits some variation across different hyperparameter values, but the impact is relatively modest. For the CIFAR10 dataset, the accuracy ranges from 95.9% to 96.3% across different affine transformation hyperparameter values, and from 96.0% to 96.3% for different color transformation hyperparameter values. Similarly, for the Flowers dataset, the accuracy ranges from 88.0% to 89.5% for affine transformation parameters and from 88.5% to 89.5% for color transformation parameters.

Importantly, ACAVP consistently outperforms the strongest baselines across all hyperparameter configurations tested. For the CIFAR10 dataset, the second-best baseline (EVP) achieves 95.87% accuracy, while ACAVP exceeds this performance regardless of the hyperparameter values selected. Similarly, for the Flowers dataset, the second-best baseline (AutoVP) achieves 82.69% accuracy, which is substantially lower than even the worst-performing ACAVP configuration.

These results demonstrate that while ACAVP's performance can be further optimized through hyperparameter tuning, the method is relatively robust to hyperparameter choices. This robustness is particularly valuable in practical applications, as it suggests that ACAVP can achieve strong performance even without extensive hyperparameter optimization. The consistent performance advantage over baseline methods across all hyperparameter configurations further validates the effectiveness of the proposed approach.

## E.4 Experiments with other models

We conducted additional experiments using the other models (SigLIP [Zhai et al., 2023], DINOv2 [Oquab et al., 2023], ViT-L/16, and ResNet101). All experiments were conducted with a single run. We set the number of training epochs to 200. For learning rate optimization, we evaluated two different learning rates: 40 and 0.1, and reported the results from the higher-performing configuration. The results are presented in Table 11. ACAVP achieves the highest accuracy across all architectures except DINOv2, where it achieves the second-highest performance. All VP methods, including ACAVP, can improve the performance of models with high zero-shot capabilities such as SigLIP. Notably, ACAVP demonstrates strong performance even with state-of-the-art vision encoders.

Next, Table 12 shows the computational overhead (the relative time compared to the inference time of each model) for each model. Since these models have shorter inference times than the CLIP ViT-B/32

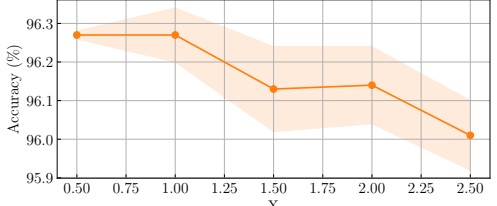

(a) AVCAVP accuracy when changing the hyperparameters of the affine transformation on the CIFAR10 dataset.

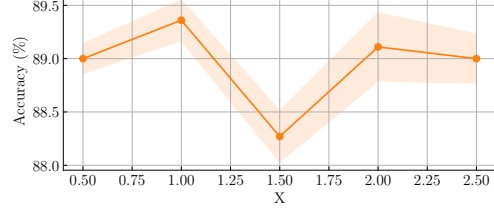

(b) AVCAVP accuracy when changing the hyperparameters of the affine transformation on the Flowers dataset.

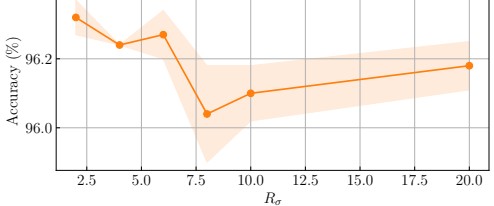

(c) AVCAVP accuracy when changing the hyperparameter of the color transformation on the CIFAR10 dataset.

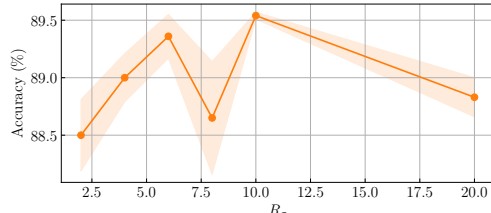

(d) AVCAVP accuracy when changing the hyperparameter of the color transformation on the Flowers dataset.

Figure 3: Performance of ACAVP with different hyperparameter values using CLIP ViT-B/32. (a) Accuracy on the CIFAR10 dataset when scaling affine transformation hyperparameters ($R_t$, $R_\theta$, $R_{sh}$) by a factor of $X$ from their default values, with the x-axis showing the value of $X$. (b) Accuracy on the Flowers dataset when scaling affine transformation hyperparameters by a factor of $X$, with the x-axis showing the value of $X$. (c) Accuracy on the CIFAR10 dataset with different values of the color transformation hyperparameter $R_\sigma$, with the x-axis showing the value of $R_\sigma$. (d) Accuracy on the Flowers dataset with different values of the color transformation hyperparameter $R_\sigma$, with the x-axis showing the value of $R_\sigma$.

Table 12: Inference time comparison across different methods using H100 GPU with batch size 500. Relative Time is normalized to the model's inference time. The parentheses indicate model architecture specifications.

| Method | ResNet101 | ViT-L/16 | DINOv2(ViT-S/14) | SigLIP(ViT-SO400M/14) |
|---|---|---|---|---|
| VP | 0.00 | 0.01 | 0.01 | 0.00 |
| EVP | 0.02 | 0.03 | 0.04 | 0.01 |
| AutoVP | 1.94 | 3.38 | 5.33 | 1.53 |
| ACAVP | 0.18 | 0.31 | 0.49 | 0.14 |
| Coordinator | 21.92 | 38.24 | 60.33 | 17.32 |

used in the main experiments, the relative times for ACAVP are slightly higher than those reported in Table 8. However, across all models, ACAVP requires less than half the inference time of the model itself. Notably, for the high-performing SigLIP, the relative time is only 0.14, indicating minimal overhead. While we have not yet pursued this direction, further optimization of the implementation could potentially reduce the computational time of VP methods even more.

## E.5 Experiments on the ImageNet dataset

The VP method primarily targets users with limited computational resources and is generally evaluated on small to medium-sized datasets, but demonstrating performance on standard benchmarks is crucial. For this purpose, we conducted additional experiments using the ImageNet dataset [Russakovsky et al., 2015]. We used CLIP ViT-B/16 and trained ACAVP for 10 epochs. The results are presented in Table 13. Although the performance gain is modest in this preliminary experiment, it demonstrates

Table 13: Classification accuracy comparison on the ImageNet dataset using CLIP ViT-B/16.

| Method | ImageNet |
|--------|----------|
| ZS     | 67.48    |
| ACAVP  | **67.64** |

Table 14: Comparison of classification accuracy with generative models. We used InstructBLIP for VP training and did not use BLIP2 during training, only during testing.

| Model | InstructBLIP | | BLIP2 | |
|-------|---------|----------|---------|----------|
| Method | CIFAR10 | CIFAR100 | CIFAR10 | CIFAR100 |
| ZS    | 88.11 | 82.41 | 58.41 | 60.65 |
| EVP   | 98.40 | 85.05 | 83.20 | 58.39 |
| TVP   | 98.78 | 83.10 | 85.15 | 62.80 |
| ACAVP | **98.85** | **88.72** | **85.32** | **66.67** |

that ACAVP can be extended to large-scale datasets like ImageNet. This result highlights the potential flexibility and scalability of our method beyond its original low-resource target scenario.

### E.6 Experiments on generative models and transferability to unseen models

Generative models are increasingly prevalent in current state-of-the-art technology, making it important to demonstrate that our method is effective against such models. We used InstructBLIP [Dai et al., 2023] as the recognition model for VP training and evaluated both InstructBLIP and BLIP2 [Li et al., 2023] on the CIFAR10 and CIFAR100 datasets. BLIP2 was not used during VP training but only during evaluation. During this experiment, we did not access BLIP2's architecture or gradient information; we used only its input-output interface. This setting replicates the typical usage scenario of cloud APIs and demonstrates the applicability of our method in actual black-box environments. The results are presented in Table 14. The results demonstrate that all VP methods are indeed effective on generative models like InstructBLIP. Notably, ACAVP achieves the highest accuracy, outperforming existing methods across both datasets. Interestingly, ACAVP exhibits superior transferability compared to TVP, even though TVP incorporates additional loss functions specifically designed to enhance transferability to models not used during training (BLIP2 in this case). This suggests that the expanded transformation space and overfitting mitigation introduced by ACAVP effectively improve VP's generalization performance and transferability.

### E.7 Experiments on the object detection and semantic segmentation tasks

We conducted experiments on the object detection and semantic segmentation tasks to evaluate the broader applicability of our approach. For the object detection task, we trained and evaluated an SSD [Liu et al., 2016] model (with ImageNet pre-trained VGG16 [Simonyan and Zisserman, 2014] backbone) on PASCAL VOC2007 [Everingham et al., 2015]. For the semantic segmentation task, we trained and evaluated a DeepLabv3 [Chen et al., 2017] model (with ImageNet pre-trained ResNet50 backbone) on PASCAL VOC2012. We jointly optimized VP and the classification heads of these models during training. The results are presented in Table 15. ACAVP demonstrates superior performance compared to the baseline VP method across both tasks. These results indicate that ACAVP's enhanced transformation space is effective beyond image classification tasks. While the absolute performance on the object detection task is relatively modest, this can be attributed to experimental constraints imposed by limited time resources, specifically the reduced number of training epochs and the use of only PASCAL VOC2007 for training (whereas standard practice typically involves training on both PASCAL VOC2007 and PASCAL VOC2012 datasets).

### E.8 Comparisons with Visual Prompt Tuning

We compare ACAVP and VPT [Jia et al., 2022] using the accuracy reported in [Wu et al., 2024]. The results are presented in Table 16. While VPT performs better on some datasets, ACAVP outperforms VPT on the majority of them, demonstrating its competitive performance.

Table 15: Performance comparison on the object detection and semantic segmentation tasks.

| Method | PASCAL VOC2007 mAP | PASCAL VOC2012 mIoU |
|---|---|---|
| ZS | 32.8 | 62.3 |
| ACAVP | **33.0** | **64.1** |

Table 16: Classification accuracy comparison of ACAVP and VPT using CLIP ViT-B/32.

| Method | CIFAR10 | CIFAR100 | CLEVR | DTD | EuroSAT | Flowers | Food | Pets | SUN | SVHN |
|---|---|---|---|---|---|---|---|---|---|---|
| VPT | 95.0 | 76.6 | 58.6 | 61.6 | 94.6 | 76.2 | **84.7** | **92.1** | **69.3** | 86.1 |
| ACAVP | **96.27** $\pm 0.07$ | **80.06** $\pm 0.25$ | **77.19** $\pm 0.17$ | **67.43** $\pm 0.83$ | **97.51** $\pm 0.09$ | **89.36** $\pm 0.19$ | 77.86 $\pm 0.09$ | 87.55 $\pm 0.02$ | 64.95 $\pm 0.10$ | **91.81** $\pm 0.18$ |

# F Limitations

While ACAVP demonstrates superior performance across diverse datasets and model architectures, it has certain limitations that should be acknowledged. First, although the computational overhead of ACAVP is relatively minimal compared to model inference (only $6\%$ of the model's computational cost), it still requires more computation than simpler VP methods such as VP and EVP. This increased computational requirement might become relevant in extremely resource-constrained environments where even small additional costs are significant.

ACAVP's transformation space, while significantly more expressive than conventional VP approaches, is still limited to affine, color, and additive transformations. More complex transformations, such as non-linear geometric transformations or advanced image processing operations, could potentially further enhance the expressive power of VP. However, incorporating such transformations would likely increase the computational overhead and potentially introduce additional training challenges.

