# OpenReview forum: "Enhancing Visual Prompting through Expanded Transformation Space and Overfitting Mitigation"
_NeurIPS.cc/2025/Conference — NeurIPS 2025 poster_

### Official Review · Reviewer_Aezh · 2025-06-02

**Clarity:** 3
**Significance:** 2
**Originality:** 3
**Rating:** 4
**Confidence:** 3

**Summary:**

While conventional VP methods achieve lower accuracy than other adaptation approaches, the paper proposes ACAVP (Affine, Color, and Additive Visual Prompting), which enhances VP’s expressive power by introducing complementary transformation operations. Here affine transformation can create task-specific prompt regions while preserving original image information, and color transformation, on the other hand, can emphasize task-relevant visual features.

**Questions:**

Please see my above concern.

**Ethical Concerns:**

["NO or VERY MINOR ethics concerns only"]

**Paper Formatting Concerns:**

I did not notice paper formatting concerns.

**Quality:**

3

**Strengths And Weaknesses:**

Strength:
1. The paper is easy to follow. By introducing the affine transformation for creating task-specific prompt regions, this approach is able to robustly improve VP's performance.

Weakness:
1. A missing area of this paper is visual prompt tuning/prefix tuning in language [1-4], which is very similar to visual prompting techniques. The authors need to clearly separate the difference between VP and VPT. Also, as stated in [5], visual prompt tuning seems to be competitive with full fine-tuning under some conditions. Does the claim "VP methods typically achieve lower accuracy than full fine-tuning" logically correct?
2. Similar to [6], the authors may need to consider involving other competitive PEFT baselines for completeness (e.g., prompt tuning, adapter)

[1] Visual prompt tuning

[2] E2VPT: An Effective and Efficient Approach for Visual Prompt Tuning

[3] Visual Fourier Prompt Tuning

[4] Aprompt: Attention prompt tuning for efficient adaptation of pre-trained language models

[5] Facing the Elephant in the Room: Visual Prompt Tuning or Full Finetuning?

[6] BlackVIP: Black-Box Visual Prompting for Robust Transfer Learning

---

> ### Author Rebuttal · Authors · 2025-07-31
>
> We thank the reviewer for this thorough review and for recognizing the clarity of our paper and the effectiveness of the proposed affine transformation in improving VP performance.
> We address your concerns below.
>
> # Weaknesses.1
> > A missing area of this paper is visual prompt tuning/prefix tuning in language [1-4], which is very similar to visual prompting techniques. The authors need to clearly separate the difference between VP and VPT.
>
> Thank you for the insightful comment.
> We agree that visual prompt tuning and prefix tuning in language domains are conceptually related to visual prompting (VP).
> As described in [1–4], VPT adapts to downstream tasks by optimizing input embeddings as well as the classification head of the model.
> In contrast, VP methods attach learnable parameters directly to the input image, and typically adjust the classification head using heuristic methods [7, 8, 9], rather than end-to-end optimization.
>
> Importantly, VP methods are often designed for settings where the architecture of the recognition model is unknown or the gradients are inaccessible, as in black-box scenarios [6].
> We will revise the paper to more clearly articulate these differences between VP and VPT, and to appropriately position our work in this context.
>
> [7]  Wu, Junyang, et al. "Unleashing the power of visual prompting at the pixel level." TMLR. 2024.
>
> [8] Chen, Aochuan, et al. "Understanding and improving visual prompting: A label-mapping perspective." Proceedings of the IEEE/CVF Conference on Computer Vision and Pattern Recognition. 2023.
>
> [9] Cai, Chengyi, et al. "Bayesian-guided label mapping for visual reprogramming." The Thirty-eighth Annual Conference on Neural Information Processing Systems. 2024.
>
>
> > Also, as stated in [5], visual prompt tuning seems to be competitive with full fine-tuning under some conditions. Does the claim "VP methods typically achieve lower accuracy than full fine-tuning" logically correct?
>
> Thank you for highlighting this interesting point.
> As shown in [5], when the pretraining and downstream datasets share similar distributions, VPT can outperform full fine-tuning (FT).
> This is consistent with our own results (Table 2), where ACAVP achieves higher accuracy than FT on datasets such as CIFAR-10 and SUN.
> We agree that the original statement "VP methods typically achieve lower accuracy than full fine-tuning" is somewhat of an overclaim.
> A more accurate description would be "VP methods often achieve lower accuracy than full fine-tuning."
> We will revise the wording accordingly in the camera-ready version.
>
>
> # Weaknesses.2
> > Similar to [6], the authors may need to consider involving other competitive PEFT baselines for completeness (e.g., prompt tuning, adapter)
>
> Thank you for pointing this out. While [6] does not include empirical accuracy comparisons with PEFT methods such as VPT or prompt tuning, it does provide a taxonomy comparing their settings (e.g., prompt location, gradient access, and input dependency).
> Our method is an extension of VP and thus shares the same setting as VP and BlackVIP.
> Nevertheless, to further clarify the relationship between VP-based methods and other PEFT approaches such as VPT and adapter tuning, we will explicitly describe these differences in the related work section of the camera-ready version.
>
> Additionally, we include below a comparison of ACAVP and VPT using the accuracy reported in [7]:
>
>
> | Method | CIFAR10          | CIFAR100         | CLEVR            | DTD              | EuroSAT          | Flowers          | Food         | Pets         | SUN          | SVHN             |
> |--------|------------------|------------------|------------------|------------------|------------------|------------------|--------------|--------------|--------------|------------------|
> | VPT    | 95               | 76.6             | 58.6             | 61.6             | 94.6             | 76.2             | **84.7**     | **92.1**     | **69.3**     | 86.1             |
> | ACAVP  | **96.27 ± 0.07** | **80.06 ± 0.25** | **77.19 ± 0.17** | **67.43 ± 0.83** | **97.51 ± 0.09** | **89.36 ± 0.19** | 77.86 ± 0.09 | 87.55 ± 0.02 | 64.95 ± 0.10 | **91.81 ± 0.18** |
>
>
> While VPT performs better on some datasets, ACAVP outperforms VPT on the majority of them, demonstrating its competitive performance.

---

> > ### Comment · Reviewer_Aezh · 2025-08-05
> >
> > I appreciate the detailed rebuttal. Please make sure that all of the discussions and experiments are included in the revision. The discussions/differences between VP and Visual prompt tuning related research are fundamental. I thus maintain my rating to this paper.

---

> > > ### Author Response · Authors · 2025-08-05
> > >
> > > Thank you for providing your comment.
> > > We sincerely hope that our rebuttal has addressed your concerns.
> > > We commit to incorporating the additional experiments conducted in our rebuttal and the clarification of the differences between VP and VPT into the camera-ready version.

---

### Official Review · Reviewer_63vY · 2025-06-21

**Clarity:** 3
**Significance:** 3
**Originality:** 3
**Rating:** 4
**Confidence:** 4

**Summary:**

This study addresses two fundamental limitations in visual prompting (VP)—namely, restricted expressive power and pronounced overfitting tendencies—by introducing ACAVP (Affine, Color, and Additive Visual Prompting), a multimodal visual prompting framework that integrates affine transformations, color transformations, and additive perturbations.​This approach fundamentally expands VP's transformation space by dynamically generating task-adaptive prompt regions through affine transformations (preserving original image information) while enhancing discriminative features via color transformations (adjusting brightness/contrast). Crucially, it systematically identifies TrivialAugment data augmentation as the optimal overfitting mitigation strategy. Extensive experimentation across 12 datasets and two architectures (ViT/ResNet50) demonstrates that ACAVP achieves state-of-the-art performance: attaining 83.18% average accuracy on CLIP-ViT (surpassing all existing VP methods and linear probing), improving robustness by 7.33% on distribution-shifted CIFAR10-C data, and maintaining VP's computational efficiency with merely 6% inference overhead. This work systematically resolves the fundamental trade-off between expressive power and generalization capability.

**Questions:**

Question 1

The paper claims ACAVP enhances visual prompt expressiveness via affine and color transformations but does not rigorously demonstrate the complementarity of these operations. How do the authors rule out redundancy—i.e., whether a single transformation alone can achieve similar or better performance?

Question 2

Does the distribution of zero-value pixels after affine transformation exhibit task adaptability? As shown in Supplementary Figure 2, affine transformation magnitudes vary significantly between datasets (CIFAR10 vs. Flowers)—does this lead to instability in masked regions?

Question 3

Why was TrivialAugment selected as the default configuration? Is its computational efficiency better aligned with the lightweight design principle of visual prompts (VPs)?

**Ethical Concerns:**

["NO or VERY MINOR ethics concerns only"]

**Final Justification:**

The authors’ rebuttal and subsequent discussions have addressed most of the concerns and provided additional clarifications and experiments regarding black-box model settings (such as cloud APIs) and larger-scale datasets. Therefore, I have decided to raise my score.

**Limitations:**

Experimental validation is inadequate and needs extension. Specifically, the study lacks comparative evaluations of more diverse backbone architectures and fails to benchmark on a broader range of mainstream datasets.

**Quality:**

3

**Strengths And Weaknesses:**

Strengths:

1.Proposes a tri-modal visual prompt framework (ACAVP) fusing affine, color, and additive transformations, which systematically addresses the intrinsic limitation of traditional visual prompt (VP) methods—their exclusive reliance on additive noise as the sole prompt form.
2.Uncovers general patterns of overfitting in VP tasks and demonstrates the universal utility of TrivialAugment for improving VP generalization, providing a critical regularization paradigm for the field.

 Weaknesses：

1.The experiments are solely conducted on two architectures CLIP ViT-B/32 and ResNet50，and computational overhead (e.g., inference time, GPU memory consumption) is only reported for CLIP ViT-B/32, lacking comparative analysis across different backbones. To support claims of generality, overhead data for additional architectures (e.g., ViT-L/14, ResNet101) should be supplemented, along with an analysis of how overhead scales with architecture complexity.

2.The ablation experiments for the three core components are incompleteFurthermore, the backbone architecture used in these ablations is not explicitly stated.  Comprehensive ablation studies including full combinations should be expanded, and the backbone corresponding to each ablation explicitly labeled to ensure conclusion traceability.

3.Experiments in Tables 5 and 6 are validated on only two datasets, providing insufficient coverage to fully demonstrate the method’s generalization ability across diverse data distributions. Most critically, results on canonical benchmarks like ImageNet are absent.

---

> ### Author Rebuttal · Authors · 2025-07-31
>
> We thank the reviewer for this thorough review and for recognizing our systematic approach to addressing VP's limitations through the tri-modal framework and our identification of overfitting patterns with effective regularization solutions.
> We address your concerns below.
>
> # Weaknesses.1
> > The experiments are solely conducted on two architectures CLIP ViT-B/32 and ResNet50，and computational overhead (e.g., inference time, GPU memory consumption) is only reported for CLIP ViT-B/32, lacking comparative analysis across different backbones. To support claims of generality, overhead data for additional architectures (e.g., ViT-L/14, ResNet101) should be supplemented, along with an analysis of how overhead scales with architecture complexity.
>
> Thank you for this important feedback.
> We conducted additional experiments using ViT-L/16 and ResNet101 (we used ViT-L/16 instead of ViT-L/14 as the latter is not available in torchvision.models) as you suggested, as well as SigLIP and DINOv2 as requested by reviewer wRed.
> The accuracy results are shown in the table in our response to reviewer wRed, and the computational overhead (relative time compared to each model's inference time) is presented in the table below.
>
> | Method      | RN101 | ViT-L/16 | DINOv2(ViT-S/14) | SigLIP(ViT-SO400M/14) |
> |-------------|-------|----------|------------------|-----------------------|
> | VP          | 0.00  | 0.01     | 0.01             | 0.00                  |
> | EVP         | 0.02  | 0.03     | 0.04             | 0.01                  |
> | AutoVP      | 1.94  | 3.38     | 5.33             | 1.53                  |
> | ACAVP       | 0.18  | 0.31     | 0.49             | 0.14                  |
> | Coordinator | 21.92 | 38.24    | 60.33            | 17.32                 |
>
>
> Since these models have shorter inference times than the CLIP ViT-B/32 used in our paper, the relative times for ACAVP are slightly higher than those reported in Table 8.
> However, across all models, ACAVP requires less than half the inference time of the model itself.
> Notably, for the high-performing SigLIP, the relative time is only 0.14, indicating minimal overhead.
> While we have not yet pursued this direction, further optimization of the implementation could potentially reduce the computational time of VP methods even more.
>
> # Weaknesses.2 & Questions.1
> > The ablation experiments for the three core components are incompleteFurthermore, the backbone architecture used in these ablations is not explicitly stated. Comprehensive ablation studies including full combinations should be expanded, and the backbone corresponding to each ablation explicitly labeled to ensure conclusion traceability.
>
> > The paper claims ACAVP enhances visual prompt expressiveness via affine and color transformations but does not rigorously demonstrate the complementarity of these operations. How do the authors rule out redundancy—i.e., whether a single transformation alone can achieve similar or better performance?
>
> Thank you for pointing out the missing specification of the model used in our ablation studies.
> We used CLIP ViT-B/32 for all ablation experiments.
> While we measured the impact of adding Affine and Color transformations to EVP in our paper, we had not evaluated the individual effects of Affine and Color transformations alone.
> We conducted additional comprehensive ablation studies to address this concern.
> The results are shown in the table below.
>
> | Transformation                    | CIFAR10      | Flowers      |
> |-----------------------------------|--------------|--------------|
> | None (ZS)                         | 88.93        | 61.63        |
> | Affine                            | 90.17 ± 0.01 | 57.42 ± 0.03 |
> | Color                             | 92.14 ± 0.06 | 62.92 ± 0.04 |
> | Affine + Color                    | 92.46 ± 0.04 | 60.58 ± 0.08 |
> | Affine + Color + Additive (ACAVP) | 96.27 ± 0.07 | 89.36 ± 0.19 |
>
> Individual transformations show varying effectiveness across datasets: Affine transformation improves performance on CIFAR10 but shows reduced performance on Flowers compared to zero-shot, while Color transformation consistently outperforms zero-shot accuracy on both datasets.
> Combining Affine and Color transformations yields further improvements.
> Furthermore, adding the traditional Additive transformation achieves the highest accuracy.
> These results demonstrate that Affine, Color, and Additive transformations have complementary relationships, and the use of these transformations is not redundant.
>
> # Weaknesses.3
> > Experiments in Tables 5 and 6 are validated on only two datasets, providing insufficient coverage to fully demonstrate the method’s generalization ability across diverse data distributions.
>
> We appreciate your valuable feedback. Due to limited computational resources, we were unable to conduct the detailed analysis (Tables 5 and 6) across all 12 datasets used in the main experiments.
> These supplementary experiments were designed to provide deeper insights into the internal workings of ACAVP, while the main experimental results (Tables 2 and 3) already verify its generalization capability across a diverse set of tasks.
> Nonetheless, we agree that additional dataset coverage would further strengthen our claims.
> We will work to expand these analyses to all 12 datasets in the camera-ready version.
>
> > Most critically, results on canonical benchmarks like ImageNet are absent.
>
> Thank you for pointing this out. While VP methods, including ours, are primarily aimed at users with limited resources and are generally evaluated on small- to medium-scale datasets, we understand the importance of demonstrating performance on canonical benchmarks.
> To that end, we performed an additional experiment on ImageNet.
> As shown below, ACAVP achieved slightly better performance than the zero-shot baseline, using CLIP ViT-B/16 fine-tuned for 10 epochs:
>
> | Method | Accuracy  |
> |--------|-----------|
> | ZS     | 67.48     |
> | ACAVP  | **67.64** |
>
> Although the performance gain is modest in this preliminary experiment, it demonstrates that ACAVP can be extended to large-scale datasets like ImageNet. This result highlights the potential flexibility and scalability of our method beyond its original low-resource target scenario.
>
>
> # Questions.2
> > Does the distribution of zero-value pixels after affine transformation exhibit task adaptability? As shown in Supplementary Figure 2, affine transformation magnitudes vary significantly between datasets (CIFAR10 vs. Flowers)—
>
> Yes, the distribution of zero-value pixels after affine transformation does exhibit task adaptability.
>
> Our dynamic mask generation mechanism (M) automatically adapts to the zero-value pixel distribution created by the learned affine transformation for each task. When the affine transformation creates larger empty regions (as in Flowers), more pixels become available for additive prompting, allowing for greater input space modification. Conversely, when smaller transformations are sufficient (as in CIFAR10), fewer zero-value pixels are created, resulting in more focused prompt application.
>
> This adaptive behavior reflects the varying adaptation requirements across tasks:
> - **Distribution-similar tasks** (ImageNet→CIFAR10): Minimal affine transformation creates small zero-value regions, indicating that subtle spatial adjustments with limited additive prompting suffice
> - **Distribution-different tasks** (ImageNet→Flowers): Larger affine transformations create extensive zero-value regions, enabling more substantial input modifications through expanded additive prompting areas
>
> > does this lead to instability in masked regions?
>
> Yes, the masked regions are inherently unstable as they vary across different training runs even with the same task and model due to the stochastic nature of the learning process.
> However, this instability in masked regions does not adversely affect task performance.
> According to the results in Tables 2 and 3, ACAVP exhibits small standard errors across most experimental settings, demonstrating that recognition performance remains stable despite the variability in mask generation.
> The key insight is that while individual mask patterns may vary between training runs, the overall adaptation capability and performance remain consistent, indicating that our method learns robust task-specific transformations regardless of the specific mask configuration.
>
> # Questions.3
> > Why was TrivialAugment selected as the default configuration? Is its computational efficiency better aligned with the lightweight design principle of visual prompts (VPs)?
>
> Thank you for the thoughtful question.
> To ensure practical usability without the overhead of policy search, we considered three representative data augmentation methods that are lightweight and easy to apply. As shown in Table 6, TrivialAugment yielded the best performance among them, and was thus selected as the default configuration.
>
> While TrivialAugment introduces some additional cost during training, it has no impact on inference-time computation. Since the key design principle of VPs lies in minimizing inference-time overhead, the use of TrivialAugment preserves the lightweight nature of our method.

---

> > ### Comment · Reviewer_63vY · 2025-08-04
> >
> > Thank you for your detailed rebuttal. Your responses have addressed most of my concerns. However, I would like to follow up on the ImageNet experiments. Specifically, when comparing with the zero-shot baseline, the performance gain achieved by ACAVP appears to be relatively small. Does this mean that ACAVP only performs well on small datasets and performs poorly on large datasets?
> >
> > I would appreciate it if the authors could provide a more detailed explanation for this observation. If the explanation is reasonable and/or additional evidence is provided, I would be happy to increase my score. In addition, it is suggested that the author can consider comparing with relevant programs of Prompt learning to highlight the progressiveness of ACAVP, such as CoOp [1], PromptSRC [2], Maple [3], or CasPL [4].
> >
> > [1] CoOp: https://github.com/KaiyangZhou/CoOp
> >
> > [2] PromptSRC: https://github.com/muzairkhattak/PromptSRC
> >
> > [3] Maple: https://github.com/muzairkhattak/multimodal-prompt-learning
> >
> > [4] CasPL: https://github.com/megvii-research/CasPL

---

> > > ### Author Response · Authors · 2025-08-04
> > >
> > > We are pleased that our rebuttal has addressed most of your concerns. We would like to provide additional clarification regarding the experimental results on the ImageNet dataset.
> > >
> > > > Does this mean that ACAVP only performs well on small datasets and performs poorly on large datasets?
> > >
> > > No, this is not the case. VP approaches, including ACAVP, perform better when there is a larger distribution gap between the pre-training dataset used for the recognition model and the downstream dataset, and when the visual variation diversity within the dataset is lower (please refer to Section 6.1 of [5]).
> > >
> > > Indeed, our experimental results support this hypothesis:
> > >
> > > - **Datasets with distant distributions**: For GTSRB (traffic signs, 18.18%→92.04% substantial improvement), SVHN (street view house numbers, 15.30%→91.81% substantial improvement), and EuroSAT (satellite images, 39.02%→97.51% substantial improvement), we achieved significant improvements regardless of dataset size.
> > >
> > > - **Datasets with similar distributions**: Conversely, for datasets containing general images similar to CLIP's pre-training data, such as Food (79.67%→77.86%) and Pets (85.96%→87.55%), the improvements were modest.
> > >
> > > Therefore, it is the distribution and diversity of the dataset, rather than its size, that influences VP performance. Since ImageNet is a general image dataset with a distribution similar to CLIP's pre-training data, ACAVP's improvement was relatively modest on this dataset.
> > >
> > > [5] Bahng, Hyojin, et al. "Exploring visual prompts for adapting large-scale models." arXiv preprint arXiv:2203.17274 (2022).
> > >
> > >
> > > > In addition, it is suggested that the author can consider comparing with relevant programs of Prompt learning to highlight the progressiveness of ACAVP, such as CoOp [1], PromptSRC [2], Maple [3], or CasPL [4].
> > >
> > > Thank you for the valuable suggestion regarding comparisons with prompt learning methods. We greatly appreciate this constructive feedback.
> > > While we cannot conduct comprehensive experimental comparisons within the remaining time of the rebuttal period, we would like to commit to including these important comparisons in the camera-ready version of our paper, should it be accepted.
> > > In the meantime, we can provide a methodological comparison that highlights the fundamental differences in problem settings:
> > > | Method | Prompt Type | Parameter Location |
> > > |--------|-------------|-------------------|
> > > | CoOp | Language | Embedding Space |
> > > | VP  | Vision | Input Space |
> > >
> > > The prompt learning methods you mentioned learn parameters in the embedding space of vision-language models, specifically optimizing text prompts within the language encoder. In contrast, VP operates purely in the visual input space without requiring access to any internal model components.
> > >
> > > This fundamental difference means that prompt learning methods cannot be applied to black-box model settings, where internal embeddings are inaccessible (such as cloud APIs or proprietary models).
> > > ACAVP, however, maintains full compatibility with black-box scenarios, which is one of the key advantages of visual prompting approaches.
> > >
> > > We believe that including both methodological analysis and comprehensive experimental comparisons in the final version will provide valuable insights into the complementary strengths of these different prompting paradigms.

---

> > > > ### Comment · Reviewer_63vY · 2025-08-06
> > > >
> > > > Thank you very much for your response. You mention that prompt learning cannot be applied to black-box model settings (such as cloud APIs), whereas ACAVP can be applied in such scenarios. However, I do not find any experiments in the paper demonstrating the application of ACAVP to cloud APIs. Could the authors further clarify this point or provide supporting evidence?

---

> ### Author Response · Authors · 2025-08-08
>
> Thank you for your thoughtful question.
>
> The scope of our paper was focused on demonstrating performance improvements through VP's extended transformation space and overfitting mitigation, and the application to black-box models was outside our original scope.
> However, we can demonstrate the applicability of ACAVP to black-box model settings through: (1) additional experimental results and (2) integration with existing research.
>
> (1) Additional Experimental Results:
>
> In the generative model experiments described in our response to reviewer wRed, we conducted experiments under settings that simulate cloud API use cases.
> Specifically, we evaluated VP trained on InstructBLIP using BLIP2, which was not used during training.
> During this evaluation, we did not access BLIP2's architecture or gradient information; we used only its input-output interface.
> This setting replicates the typical usage scenario of cloud APIs and demonstrates the applicability of our method in actual black-box environments.
> The experimental results show that ACAVP outperforms existing methods even in such black-box settings, with particularly notable improvements in transferability to BLIP2.
> Specifically, ACAVP improves BLIP2's accuracy from 82.41% to 88.72% on the CIFAR10 dataset and from 60.65% to 66.67% on the CIFAR100 dataset.
>
> (2) Integration with Existing Research:
>
> Studies such as [6-8] focus on applying VP to black-box model settings.
> These studies primarily focus on training methods for VP from black-box models and are independent of specific VP architectures or shapes.
> Since ACAVP's improvements (extended transformation space and overfitting mitigation) are orthogonal to these training methodologies, our approach can be readily integrated with existing black-box VP training frameworks.
>
> [6] Oh, Changdae, et al. "Blackvip: Black-box visual prompting for robust transfer learning." Proceedings of the IEEE/CVF Conference on Computer Vision and Pattern Recognition. 2023.
>
> [7] Tsai, Yun-Yun, Pin-Yu Chen, and Tsung-Yi Ho. "Transfer learning without knowing: Reprogramming black-box machine learning models with scarce data and limited resources." International Conference on Machine Learning. PMLR, 2020.
>
> [8] Cho, Wonwoo, et al. "Training Spatial-Frequency Visual Prompts and Probabilistic Clusters for Accurate Black-Box Transfer Learning." Proceedings of the 32nd ACM International Conference on Multimedia. 2024.
>
>
> We hope this response addresses your remaining concerns.

---

> > ### Comment · Reviewer_63vY · 2025-08-08
> >
> > Thank you for the authors’ latest response, which has addressed the concerns raised above. I look forward to seeing additional ablation studies conducted under black-box model settings (such as cloud APIs). Accordingly, I have decided to raise my score to 4.

---

> > > ### Author Response · Authors · 2025-08-08
> > >
> > > Thank you very much for raising your score to 4.
> > > We greatly appreciate your constructive feedback, which has significantly improved our paper.

---

### Official Review · Reviewer_kCxW · 2025-07-02

**Clarity:** 3
**Significance:** 2
**Originality:** 3
**Rating:** 4
**Confidence:** 4

**Summary:**

This paper proposes a novel visual prompting method, ACAVP, which leverages learnable affine transformations, color adjustments, and edge pixel prompts on the input image to achieve finetuning of black-box models. Experimental results demonstrate the superiority of this method over existing visual prompting approaches.

**Questions:**

See above

**Ethical Concerns:**

["NO or VERY MINOR ethics concerns only"]

**Final Justification:**

The author resolved my issue, and I will keep my rating.

**Paper Formatting Concerns:**

The paper does not follow the correct citation format required by the NeurIPS template.

**Quality:**

3

**Strengths And Weaknesses:**

Strengths
+ The paper proposes a simple yet effective method, and the experimental results are convincing.
+ The paper also explores the overfitting issues in existing VP methods and provides an effective solution.

Weaknesses
+ My main concern lies in the value of VP as a general methodology. From the experimental results in the paper, it is evident that VP works well for models like CLIP, which already perform strongly in zero-shot settings. However, for models pre-trained on ImageNet, the performance gap between VP and full finetuning is substantial, rendering VP far from usable in practice. This suggests that the capabilities of VP may be limited to making small adjustments to already strong models. For models that do not support zero-shot inference to begin with, VP may not be a promising direction.

---

> ### Author Rebuttal · Authors · 2025-07-31
>
> We thank the reviewer for this constructive review and for recognizing the effectiveness of our method and our contribution in addressing overfitting issues in visual prompting.
> We address your concerns below.
>
> # Weaknesses
> > My main concern lies in the value of VP as a general methodology. From the experimental results in the paper, it is evident that VP works well for models like CLIP, which already perform strongly in zero-shot settings. However, for models pre-trained on ImageNet, the performance gap between VP and full finetuning is substantial, rendering VP far from usable in practice. This suggests that the capabilities of VP may be limited to making small adjustments to already strong models. For models that do not support zero-shot inference to begin with, VP may not be a promising direction.
>
> We appreciate this insightful observation about VP's performance characteristics across different model types.
>
> We acknowledge that VP indeed achieves higher accuracy when applied to models with strong zero-shot capabilities like CLIP compared to ImageNet pre-trained models, as you correctly point out.
> However, we believe this does not diminish VP's practical utility for several reasons.
>
> Most importantly, in current practice, CLIP and similar vision-language models have become the predominant choice for vision tasks due to their superior zero-shot capabilities and versatility.
> While CLIP models have slightly higher inference costs due to text encoding, their significantly superior performance and zero-shot capabilities make them the preferred choice for practical applications.
> From this perspective, the performance gap on ImageNet pre-trained models is less concerning in practice, as practitioners typically prefer to fine-tune stronger foundation models like CLIP rather than ImageNet pre-trained models.
>
> Furthermore, our results demonstrate that VP is not limited to small adjustments to already strong models. For instance, on specialized datasets like GTSRB and SVHN, CLIP's zero-shot performance is quite low (18.18% and 15.30% respectively), yet VP methods achieve substantial improvements to over 90% accuracy (92.04% and 91.81% with ACAVP). These dramatic performance gains—improvements of over 70 percentage points—clearly show that VP can enable significant task adaptation rather than merely fine-tuning already capable models.
>
> Additionally, VP possesses a unique characteristic that distinguishes it from other parameter-efficient fine-tuning methods: its applicability to black-box models where the architecture is unknown and gradients are inaccessible.
> This capability enables VP to adapt state-of-the-art multimodal large language models provided as APIs to downstream tasks, which is impossible with traditional fine-tuning approaches.
>
> While we included ImageNet pre-trained model experiments for comprehensive evaluation, we believe the practical significance lies in VP's performance on modern foundation models like CLIP, where our method demonstrates clear advantages.
> Therefore, we believe VP represents a promising research direction that addresses scenarios where conventional adaptation methods are not applicable, particularly in the era of increasingly powerful foundation models distributed through API services.

---

> ### Comment · Reviewer_kCxW · 2025-08-05
>
> Thank you for your response. I will keep my score.

---

> > ### Author Response · Authors · 2025-08-07
> >
> > Thank you for your reply.
> > We hope our response has resolved your concerns.

---

### Official Review · Reviewer_wRed · 2025-07-02

**Clarity:** 3
**Significance:** 2
**Originality:** 1
**Rating:** 4
**Confidence:** 4

**Summary:**

The authors propose ACAVP to improve visual prompting by introducing complementary (i.e., affine, color, and additive) transformation operations. They also use TrivialAugment to mitigate overfitting problems in VP. The authors evaluate their model on CLIP and Resnet-50 on 12 image classification datasets as well as robustness to distribution shifts on CIFAR10-C and CIFAR100-C datasets.

**Questions:**

1. I wonder if the authors have tried tasks other than image classification. There are other vision tasks such as segmentation, object detection, etc. Besides vision tasks, there are multi-modal tasks as well such as VQA.
2. Since CLIP and ResNet seem a bit outdated, I think the authors can try more modern vision models such as SigLIP, DINOv2 and also other larger vision models to see if the method can scale up. Besides discriminative models, it would be interesting to see if the method can also generalize to generative models such as Qwen.
3. It would be better if the authors can give a better explanation of why specifically choosing affine/color transformation and not other transformation techniques to improve the expressivity.

**Ethical Concerns:**

["NO or VERY MINOR ethics concerns only"]

**Final Justification:**

The authors provided additional experiments on more backbones and tasks, especially generative models. It would be better if the authors can include more SOTA models such as qwen and more tasks such as VQA. Therefore, I am updating my score to 4.

**Limitations:**

Yes.

**Paper Formatting Concerns:**

No.

**Quality:**

3

**Strengths And Weaknesses:**

# Strengths
1. The paper is well-written and easy to follow.
2. The method is simple but effective.
3. The paper is technically sound. The authors make several claims (the method can improve expressivity and mitigate overfitting, and is lightweight) and provide with corresponding experiments (SOTA results on image classification datasets, robustness to distribution shifts, computational efficiency) to support the claims.
4. The authors provide theoretical analysis along with good empirical performance.

# Weaknesses
1. The evaluation setting is very limited since the authors only evaluate their model on image classification tasks.
2. The backbones are also limited since the authors only use CLIP and ResNet which seem a bit outdated give new vision encoders recently.
3. The method seems lack novelty, since in other areas people have also tried affine/color transformations.

---

> ### Author Rebuttal · Authors · 2025-07-31
>
> We sincerely thank the reviewer for the constructive feedback and for recognizing the technical soundness, clarity, and comprehensive experimental validation of our work.
> We address your valuable concerns below.
>
> # Weaknesses.1 & Questions.1
> > The evaluation setting is very limited since the authors only evaluate their model on image classification tasks.
>
> > I wonder if the authors have tried tasks other than image classification. There are other vision tasks such as segmentation, object detection, etc. Besides vision tasks, there are multi-modal tasks as well such as VQA.
>
> Thank you for this valuable feedback.
> We conducted additional experiments on object detection and semantic segmentation tasks to evaluate the broader applicability of our approach.
> For the object detection task, we trained and evaluated an SSD model (with ImageNet pre-trained VGG16 backbone) on PASCAL VOC2007.
> For the semantic segmentation task, we trained and evaluated a DeepLabv3 model (with ImageNet pre-trained ResNet50 backbone) on PASCAL VOC2012.
> We jointly optimized VP and the classification heads of these models during training.
> The results are presented in the table below.
>
> | Method | PASCAL VOC2007 mAP | PASCAL VOC2012 mIoU |
> |--------|--------------------|---------------------|
> | VP     | 32.8               | 62.3                |
> | ACAVP  | **33.0**           | **64.1**            |
>
> ACAVP demonstrates superior performance compared to the baseline VP method across both tasks.
> These results indicate that ACAVP's enhanced transformation space is effective beyond image classification tasks.
> While the absolute performance on the object detection task is relatively modest, this can be attributed to experimental constraints imposed by limited time resources, specifically the reduced number of training epochs and the use of only PASCAL VOC2007 for training (whereas standard practice typically involves training on both PASCAL VOC2007 and PASCAL VOC2012 datasets).
>
> # Weaknesses.2 & Questions.2
> > The backbones are also limited since the authors only use CLIP and ResNet which seem a bit outdated give new vision encoders recently.
>
> > Since CLIP and ResNet seem a bit outdated, I think the authors can try more modern vision models such as SigLIP, DINOv2 and also other larger vision models to see if the method can scale up. Besides discriminative models, it would be interesting to see if the method can also generalize to generative models such as Qwen.
>
> Thank you for your suggestion.
> We conducted additional experiments using the modern vision models you suggested (SigLIP, DINOv2) as well as ViT-L/16 and ResNet101 recommended by reviewer 63vY (we used ViT-L/16 instead of ViT-L/14 as the latter is not available in torchvision.models).
> The results are presented in the table below.
>
> | Model  | RN101†    |           | ViT-L/16† |           | DINOv2(ViT-S/14)† |           | SigLIP(ViT-SO400M/14) |           | Average     |
> |--------|-----------|-----------|-----------|-----------|-------------------|-----------|-----------------------|-----------|-------------|
> | Method | CIFAR10   | Flowers   | CIFAR10   | Flowers   | CIFAR10           | Flowers   | CIFAR10               | Flowers   |             |
> | ZS     | 57.32     | 5.156     | 69.49     | 4.953     | 72.35             | 8.892     | 95.3                  | 88.957    | 50.30225    |
> | VP     | 67.44     | 10.72     | 90.69     | 13.6      | 81.63             | 24.69     | 96.41                 | 97        | 60.2725     |
> | EVP    | 72.86     | 13.52     | 94.96     | 27.49     | 91.63             | 10.23     | 98.76                 | 98.66     | 63.51375    |
> | AutoVP | 68.82     | 13.2      | 94.75     | 25.74     | **94.25**         | **43.85** | 98.72                 | 97.48     | 67.10125    |
> | ACAVP  | **75.99** | **17.66** | **95.38** | **39.71** | 93.62             | 38.49     | **98.83**             | **98.74** | **69.8025** |
>
> † indicates the use of output mapping proposed in EVP.
> ACAVP achieves the highest accuracy across all architectures except DINOv2, where it achieves the second-highest performance.
> All VP methods, including ACAVP, can improve the performance of models with high zero-shot capabilities such as SigLIP.
> Notably, ACAVP demonstrates strong performance even with state-of-the-art vision encoders.
>
> Regarding experiments with generative models such as Qwen, we were unable to complete these within the limited rebuttal timeframe.
> Recent work by Zhang et al. [1] has demonstrated that visual prompting can be successfully applied to multimodal large language models, including generative models like MiniGPT-4, showing consistent improvements across various vision-language tasks.
> Given that ACAVP extends the transformation space of traditional VP through affine and color transformations, we believe that ACAVP would similarly benefit generative models.
>
> [1] Zhang, Yichi, et al. "Exploring the transferability of visual prompting for multimodal large language models." Proceedings of the IEEE/CVF Conference on Computer Vision and Pattern Recognition. 2024.
>
> # Weaknesses.3 & Questions.3
> > The method seems lack novelty, since in other areas people have also tried affine/color transformations.
>
> > It would be better if the authors can give a better explanation of why specifically choosing affine/color transformation and not other transformation techniques to improve the expressivity.
>
> Thank you for raising this important point about novelty and transformation choice.
> We acknowledge that affine and color transformations are well-established techniques in computer vision. However, our contribution lies not in the individual techniques themselves, but in their strategic integration into the visual prompting paradigm and the systematic analysis of their effectiveness in this specific context.
> Our choice of affine and color transformations was motivated by several key considerations:
> (1) These transformations are differentiable and can be seamlessly integrated into gradient-based optimization, unlike many other transformation techniques.
> (2)They provide fundamentally different transformation capabilities compared to traditional additive noise in VP—affine transformations modify spatial relationships while color transformations adjust feature intensities, creating a more expressive transformation space.
> (3) Both transformations maintain VP's core advantage of minimal inference overhead, unlike more complex alternatives such as neural network-based transformations.
> (4) As shown in our analysis (Appendix B), this combination provably reduces approximation error compared to existing VP methods.
>
> While the individual transformations are not novel, their principled combination, theoretical analysis, and empirical validation in the VP context represent our key contributions. This approach has enabled ACAVP to achieve state-of-the-art performance while maintaining the fundamental advantages of visual prompting.

---

> > ### Comment · Reviewer_wRed · 2025-08-04
> >
> > Thanks for the authors' thoughtful rebuttal. My concerns are mostly addressed. The authors provided additional experiments on more backbones and tasks, which I appreciate. However, they were unable to include experiments on generative models, which limits the demonstrated applicability of the method. While I acknowledge the method’s effectiveness on image classification, object detection, and semantic segmentation tasks, the lack of evaluation on generative models, which are now much more widely used, remains a concern. Therefore, I am updating my score from 2 to 3.

---

### Note · Authors · 2025-08-12

We are deeply grateful to all reviewers for their thoughtful and constructive feedback, which has enabled us to significantly strengthen our work.
Throughout the rebuttal period, we have conducted comprehensive additional experiments to address the concerns raised.
Our additional experimental evaluation includes the following components:

- Experiments on object detection and segmentation tasks to demonstrate broader applicability.
- Accuracy comparisons and computational overhead measurements across diverse pre-trained models and architectures (ResNet-101, ViT-L/16, DINOv2, SigLIP).
- Evaluation on generative models (InstructBLIP).
- Black-box model experiments simulating cloud API scenarios where only BLIP2 inputs and outputs are accessible without gradient information.
- Detailed ablation studies examining the contributions of affine and color transformations,
- Evaluation on ImageNet to establish performance on a large-scale dataset.
- Comparisons with Visual Prompt Tuning (VPT).

These extensive additional experiments have allowed us to address most of the reviewers' concerns and provide comprehensive evidence supporting our method's effectiveness and generalizability.

Thank you for your time and valuable feedback.

---

### Decision · Program_Chairs · 2025-09-17

**Decision:**

Accept (poster)

**Comment:**

This paper aims to enhance visual prompting. Specifically, it contains three main components: an affine transformation to create task-specific prompt regions, a color transformation to amplify salient features, and TrivialAugment for data augmentation. Empirical results are provided across a range of benchmarks to support its effectiveness.

Overall, the reviewers agreed that the strategy is clearly motivated and supported by strong empirical results. However, they also raised several concerns: 1) more evaluations beyond image classification should be included; 2) other competitive PEFT baselines should be considered; 3) more comprehrensive ablations are needed, such as isolating the performance gains from the affine transformation alone and testing under black-box settings; and 4) the novelty contribution should be more carefully and clearly articulated, as each individual component itself is not technical novel.

The rebuttal is considered, which addresses most of these concerns. As the result, all reviewers uniamously agree to accept this submission. For the final version, please ensure that all promised revisions from the rebuttal are carefully incorporated to further strengthen the paper.